# Impact of swell waves on atmospheric surface turbulence: Wave-turbulence decomposition methods

Mostafa Bakhoday Paskyabi[1]

[1]Geophysical Institute, Universtiy of Bergen, Bergen, Norway, and Bergen Offshore Wind Centre, Bergen, Norway

**Correspondence:** Mostafa Bakhoday Paskyabi (Mostafa.Bakhoday-Paskyabi@uib.no)

**Abstract.** To study turbulence properties, specifically vertical momentum fluxes during swell wave conditions, I investigate the impact of waves on the power spectrum and spectral coherence of turbulent wind across various spatial and temporal scales. I propose and apply a wave-turbulence decomposition method to split high-frequency surface wind data into distinct wind and wave components. Under the assumption of frozen turbulence, this method substitutes an empirically fitted spectrum for the observed/modelled wind spectrum within the wave-affected frequency range. I proceed to estimate time series of waves and turbulence through this decomposition technique. Using a few days of sonic anemometer wind measurements at 15 m height from June 20 to 26, 2015, the upward momentum transfer could be observed during high-steady ($\sim$ 7 m/s) and decaying wind conditions. During the high and decaying winds, the atmospheric stability changes between unstable and stable conditions, blurring the wave signals due to the thermally/mechanically generated turbulence. The vertical wind spectra from selected episodes within the study period, acting as benchmarks, offer detailed insights on how waves affect energy elevation within the wave frequency band during low winds, old sea, and stable boundary layer conditions. These spectra also facilitate an effective performance assessment of the proposed decomposition method. Additionally, using a theoretical model derived from sonic anemometer measurements at heights of 15 m and 20 m above the mean sea level, I parameterize the wave-contaminated coherence function, allowing for the synthetic generation of turbulent fluctuation spectra within the wave frequency band.

## 1 Introduction

Over the last several decades, a large number of laboratory studies and field experiments have shown modulation of turbulent momentum fluxes across a layer, so-called Wave Boundary Layer (WBL), on both sides of the air-sea interface (Chalikov, 1995; Rieder et al., 1994). The height of this sublayer, for example in the atmosphere, is approximately few metres and in the order of the significant wave height. The atmospheric WBL is then limited from bellow by the air-sea interface and at the top by the atmospheric surface layer in which the Monin–Obukhov Similarity Theory (MOST) is applied. Stratification may have direct and indirect impacts on the wind–wave interaction within the WBL (Semedo et al., 2009). However, such effects depend on the height of the WBL with respect to the height of the dynamic sublayer, wave age, and the relative angle between wind and wave directions. Within the WBL, particularly under the influence of swell waves with low to moderate wind speeds, MOST or the logarithmic law should not be applied in order to estimate the drag coefficient or roughness length, and the wind profiles show a jet at top of the WBL (Chalikov and Rainchik, 2011). This is because waves excite perturbations in this

sublayer in addition to contributions from the buoyancy and shear productions. Although studies over the last few decades have significantly improved understanding of turbulent flows above the surface gravity waves, detailed knowledge of the WBL and its interaction with atmospheric turbulence under varying forcing conditions are key in better understanding of turbulent processes and enhancing the accuracy of turbulent closure schemes used in oceanic, atmospheric, and climate models.

Fast-propagating swell waves generate wave-coherent structures in the WBL, and challenge the widely accepted Monin-Obukhov scaling (Högström et al., 2013; Smedman et al., 2003; Semedo et al., 2009; Rieder and Smith, 1998). Wind velocity fluctuations are influenced by both turbulence and wave orbital motions, potentially leading to overestimated turbulence parameters due to significant wave-phase-dependent modulation of airflow. To mitigate this issue, various methods, such as phase averaging, linear transformation (Veron et al., 2008; Grare et al., 2013; Buckley and Veron, 2017), and orthogonal projection of

wind onto the Hilbert space (Hristov et al., 2003; Wu et al., 2018) have been proposed and used for decomposing the turbulent velocity fields from instantaneous measurements of wind within the WBL. Many of these techniques rely on complex cross-spectra between horizontal and vertical air velocity fluctuations, and sea surface elevation to isolate the direct wave influence (Grare et al., 2013). It is worth mentioning that understanding the spatial coherence and the separation of wave-coherent structures from the wind measurements is crucial for generating turbulence data in structural load analysis models, wind turbine and

farm control implementations, and simulating wind-wave interactions during swell waves (Bakhoday-Paskyabi et al., 2022).

In this study, I utilize a set of near-surface wind and wave measurements collected during OBLEX-F1 campaign from the FINO1 offshore meteorological mast in June 2015. Based on these data, I explore the turbulent structures within the WBL, and the interactions of waves and wind stress over different atmospheric stability and sea-state conditions. I further calculate the swell-induced momentum fluxes from both available theories and high frequency observational data. Specifically, the approach

in this paper aims to unfold following key aspects:

  – Identifying and assessing swell-related wind-wave imprints on atmospheric velocity and two-point coherence structures during specific swell-dominated conditions (proposing a theoretical model for representing wave-induced coherence).

  – Removing wave-induced peak effects from wind velocity spectra using a spectral technique and reconstructing turbulence and wave time series from the wave-affected sonic measurements, during mostly stable atmospheric conditions.

This paper is organized as follows. Section 2 will briefly explain the coherence spectrum within the wave-affected frequency band, the wave-turbulence decomposition, and the estimation of wave-induced momentum fluxes. Section 3 explains measurements of wind and waves at FINO1 offshore met-mast, and Section 4 describes the verification results of the suggested methods. Section 5 provides a brief discussion and summary of the work.

## 2  Methods

In the presence of swell waves, the total wind $\boldsymbol{u} = (u, v, w)$ is linearly decomposed into the mean $\bar{\boldsymbol{u}}$, the turbulent $\boldsymbol{u}'$, and the wave-induced perturbation $\tilde{\boldsymbol{u}}$ as

$$\boldsymbol{u} = \bar{\boldsymbol{u}} + \boldsymbol{u}' + \tilde{\boldsymbol{u}}, \tag{1}$$

where $u, v,$ and $w$ are horizontal and vertical velocity components respectively. Based on this decomposition in this section, the WBL is studied through the wave-turbulence decomposition. Estimation of wave-induced motions $\tilde{\boldsymbol{u}} = (\tilde{u}, \tilde{v}, \tilde{w})$ in Eq. (1) presents theoretical challenges, and it becomes even more intricate when dealing with measurements of wind over an undulating air-sea interface. To underscore the complexity in a realistic scenario, waves are often modelled as a continuous spectrum of monochromatic waves, e.g. refer to a finite expansion in Eq. (11). Each wave component may induce airflow perturbations traveling at its phase speed. This necessitates employing a space-time Fourier transform to differentiate components moving at the flow speed from those traveling at the wave speed (Ayet and Chapron, 2022). This multiscale wind-wave coupling, mediated by wave-coherent motions, is a responsible mechanism for variations of turbulent characteristics over the swell waves. For instance using Eq. (1), the total wind stress vector over the wavy surface is given as follows

$$\boldsymbol{\tau} = \boldsymbol{\tau}' + \tilde{\boldsymbol{\tau}}, \tag{2}$$

where $\boldsymbol{\tau}' = -\rho_a(\overline{u'w'}, \overline{v'w'})$ is the turbulent stress and $\tilde{\boldsymbol{\tau}} = -\rho_a(\overline{\tilde{u}\tilde{w}}, \overline{\tilde{v}\tilde{w}})$ denotes the wave-induced stress. Here, $\rho_a$ indicates the air density. The total wind stress in Eq. (2) can be determined either through high-frequency measurements using the eddy covariance technique, to calculate the observed $\boldsymbol{\tau}'$ and $\tilde{\boldsymbol{\tau}}$, or by employing a bulk formula such as the one provided by COARE3.6 (Edson et al., 2013).

## 2.1 Wind-wave decomposition

To decompose wind and wave signals (or wave-turbulence decomposition), the energy spectrum of each velocity component in the inertial range at wavenumbers fairly above and below the wave band can be fitted using following the 1D Kaimal wavenumber spectrum:

$$\frac{kF_{\beta\beta}(k)}{\sigma_\beta^2} = \frac{A(k/k_{0\beta})}{1 + (k/k_{0\beta})^{5/3}}, \tag{3}$$

where $k$ denotes wavenumber, $\beta = u, v, w$, $A = 5\sin(3\pi/5)/(6\pi)$ is a constant, and $k_{0\beta}$ and $\sigma_\beta$ are two adjustable parameters describing the roll-off wavenumber (the length scales of turbulent eddies in the energy-containing subrange) and the standard deviation of $\beta$, respectively (Gerbi et al., 2009; Bakhoday-Paskyabi, 2019). Here, I perform a two-parameter least squares fit of Eq. (3) to our observations. This process allows us to estimate $k_{0\beta}$ and $\sigma_\beta$, which respectively characterize the variance and spatial scale of the energy-containing eddies. Wavenumber spectrum is then converted to frequency scales by invoking Taylor's frozen turbulence hypothesis, $k = \omega/\bar{u}$ where $\omega = 2\pi f$ and $\bar{u}$ is the mean (advection) wind speed:

$$df E_{\beta\beta}(f) = dk F_{\beta\beta}(k). \tag{4}$$

Here $E_{\beta\beta}(f)$ is the frequency spectrum of the wind $\beta$-component, and the derivative $dk/df$ is estimated by the use of the wave dispersion relation, see Appendix A. I apply then a two-parameter least squares fitting of the model spectrum in Eq. (3) to the measured spectrum. For fitting (in log–log space to ensure equal weight is given to all parts of the model fit), the wave-affected band is first determined as $[0.6k_p, k_p + 0.1]$ (or $[0.6f_p, f_p + 0.1]$) at which $k_p$ ($f_p$) denotes the peak wavenumber (frequency) measured from the recorded wave peak period, $T_p$. This interval is determined through a trail-and-error process using all datasets employed in this study, providing a reliable estimate of the frequency band for the entire campaign dataset too. The energy spectrum is then divided into three bands: below-wave-band ($k < 0.6k_p$), wave-band, and above-wave-band ($k > k_p + 0.1$) parts, see Fig. 1 and refer to Appendix 3 of Bakhoday-Paskyabi (2019). After discarding the wave-band, the Kaimal spectrum Eq. (3) is fitted over below and above wave-band wavenumbers and replace the wave-induced bump by the fitted curve. The wave induced spectrum is then estimated as follows

$$E_{\tilde{\beta}\tilde{\beta}}(f) = E_{\beta\beta}(f) - E_{\beta'\beta'}(f). \tag{5}$$

To estimate time series of turbulence and wave components in Eq. (5), I set $\boldsymbol{u} \leftarrow \boldsymbol{u} - \bar{\boldsymbol{u}}$ and transform Eq. (1) into the Fourier space in terms of Fourier coefficients of wind $U_j, V_j, W_j$ and waves $\tilde{U}_j, \tilde{V}_j, \tilde{W}_j$, for example $E_{uu} = |U_j|$ and $E_{\tilde{u}\tilde{u}} = |\tilde{U}_j|$. These Fourier coefficients are expressed in phasor notation as follows (Bricker and Monismith, 2007):

$$U_j = |U_j|e^{i\angle U_j}, \tag{6}$$
$$\tilde{U}_j = |\tilde{U}_j|e^{i\angle \tilde{U}_j}, \tag{7}$$

where $\angle$ denotes the phase operator, $i = \sqrt{-1}$ is the imaginary unit, and $|\cdot|$ represents the magnitude operator. By taking inverse Fourier transformation of the above two-sided equations, the time series of turbulent and wave velocities are calculated (see Eqs. 10 and 11).

## 2.2 Wind-wave interaction: Coherence and synthetic turbulence

The undulating surface of the ocean, as previously discussed, produces wave-coherent perturbations in the velocity (and pressure) fields. This has the potential to exert a dominant influence on turbulent properties within the WBL. In the case of low wind speeds and when vertical separation distances (for two-point turbulence problem) remain within typical turbulence length scales, turbulence can be considered frozen, allowing the application of Taylor's hypothesis using representations that may however deviate significantly from those in the existing literature. Moreover, the statistics of spatial structures in microscale and the similarity of flow motions across different scales, as measured by the coherence between spatially separated data, may be influenced in the presence of wavy surface.

This study proposes a theoretical model for generating a wave-correlated wind field (or the wave-affected turbulence) in both time and frequency domains. The essence of the model lies in representation of a coherence function for the fluctuating wind velocity by accounting for the impacts of surface wave processes. The model helps also to effectively isolate the wave contributions from the wind fluctuation signal, i.e. Eq. (9).

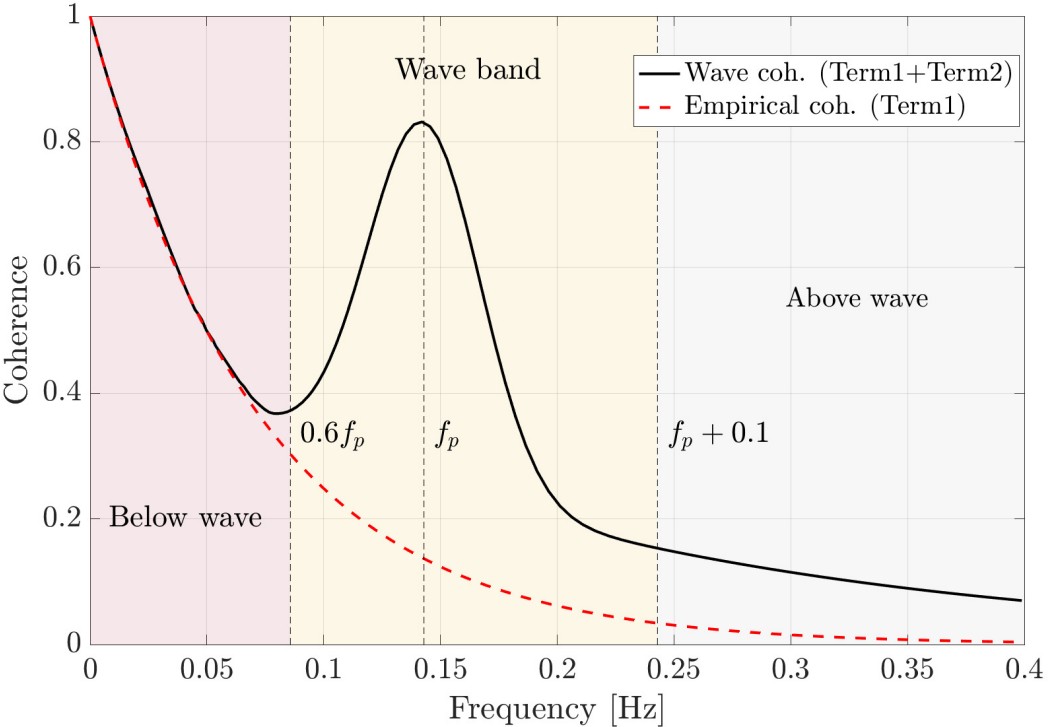

**Figure 1.** Comparing the coherence of the vertical velocity component of wind turbulence using the Davenport empirical model (Term1) to the wind-wave coherence (depicted by the black line) as shown by combining Term1 and Term2 in Eq. (8) for all frequencies $f \geq 0.6 f_p$. Here, we exclude the undulatory tail effects for $f \geq f_p + 0.1$, as presented in Term3. Here $\bar{u} = 1$ m/s, $H_s = 0.8$ m, $T_p = 8$ s, $A_1 = 0.5$, $C_2 = 0.7$, $\beta_2 = 0.09$, $A_2 = 0.6$, and $A_3 = 0.1$ for the above-the-wave frequencies (the gray-colored area).

By relying on the Davenport representation of the coherence (Davenport, 1961), I suggest following general representation for the vertical wave coherence formulation at the separation distance of $\Delta z$:

$$\gamma(\Delta z, z, f) = \overbrace{A_1 \exp\left(-C_1 \frac{f \Delta z}{\bar{u}}\right)}^{\text{Term1}} + \overbrace{A_2 \exp\left(-C_2 \lambda_w \frac{(f - f_p)^2 z}{\beta_2 [(H_s/T_p)^2 + \bar{u}^2]}\right) \cos\left[\frac{2\pi f z}{\beta_1 H_s/Tp} + \alpha\phi\right]}^{\text{Term2}} + \overbrace{A_3}^{\text{Term4}}, \tag{8}$$

where $C_1$ is a dimensionless decay parameter, $C_2$, $A_1$, $A_2$, $A_3$, $\beta_1$ and $\beta_2$ are empirically determined coefficients, through fitting with observed coherence (i.e. Eq. 12), requiring a minimum of two high-frequency wind measurements at two heights

with an appropriate separation distance. The variable $z$ represents the average height over the sea surface for which coherence is calculated (for the vertical attenuation of wave effects). Oscillations of the wave component are controlled by setting a small value for the random phase-shift of $\phi$ (i.e. $\alpha \sim 0.1$) in Term3. Here, $f$ is frequency, $\bar{u}$ denotes the mean wind speed, and $\lambda_w$ is the wave length determined by utilising the dispersion relation (Appendix A) and values of wave peak period, $T_p$, and the

significant wave height, $H_s$. In Eq. (8), Term1 represents the original Davenport empirical model of the coherence without wave disturbances for the entire frequency range (the dashed red curve in Fig. 1). The separation distances (heights) are chosen to be within the maximum range where the Taylor hypothesis remains valid. Term2 together with Term1 is used in the wave affected frequency band to represent the wind and wave coherence. The coherence in this band decreases with height $z$ above the mean surface level, and depends on $H_s$, $T_p$, wavelength, and $\bar{u}$ (the yellow area in Fig. 1). To address the oscillatory behavior in the tail of coherence spectrum, observed in the data, I incorporate sum of Term3 and Term4. It is important to note that this representation does not account for the effects of wind and wave misalignment. Furthermore, this figure does not show the effects of Term3 and the undulating behavior of Term3 can be seen in the shaded regions of Fig. 6a and b.

By estimating the squared coherence, $\gamma^2$, between vertical wind $w$ and wave elevation $\eta$ in the frequency domain, the spectral density of the wave induced and turbulence fluctuations at the height of $z$ are given according to Eq. (5) of Rieder et al. (1994):

$$E_{w'w'}(f) = (1 - \gamma^2)E_{ww}, \quad \text{and} \quad E_{\tilde{w}\tilde{w}}(f) = \gamma^2 E_{ww}. \tag{9}$$

The random realizations of the wind and wave time series for vertical velocity are then estimated according to

$$w'(t) = \Sigma_i \sqrt{2E_{w'w'}(f)\Delta f}[r_i^{(1)}\cos(\omega t) + r_i^{(2)}\sin(\omega t)], \tag{10}$$

$$\tilde{w}(t) = \Sigma_i \sqrt{2E_{\tilde{w}\tilde{w}}(f)\Delta f}[q_i^{(1)}\cos(\omega t) + q_i^{(2)}\sin(\omega t)], \tag{11}$$

where $r_i^{(1)}, r_i^{(2)}, q_i^{(1)}$ and $q_i^{(2)}$ are normal random numbers. In the synthetic time series, we determine the bulk wave parameters for the wind-sea condition using $H_s = 0.0248|U_{10}|^2$ and $T_p = 0.729|U_{10}|$ (Carter, 1982) for simplicity and a clearer conceptual visualization, here $U_{10}$ indicates the wind speed at 10 m height. It is important to note that while Eq. (9) can also be employed for wave-turbulence decomposition using two-point coherence data (see the black curves in Figs. 6c and d), our focus in this study will be solely on the spectral technique presented in the next subsection. Because the proposed method relies exclusively on high-frequency sonic data at a single height.

The observed coherence of vertical velocities is determined using the following relationship:

$$\gamma(z_1, z_2, f) = \frac{|\text{Co}_{z_1 z_2}(f)|}{\sqrt{E_{w'w'}^{z_1}(f)E_{w'w'}^{z_2}(f)}}, \tag{12}$$

where $E_{w'w'}^{z_1}(f)$ and $E_{w'w'}^{z_2}(f)$ are the the power spectral density at heights $z_1$ and $z_2$, respectively. $\text{Co}_{z_1 z_2}(f)$ denotes the two-point cross-power spectral density at heights $z_1$ and $z_2$.

## 2.3 Air-sea momentum flux

The wave boundary layer is a region where the non-static pressure distribution becomes apparent, with a height of impact corresponding to several significant wave heights. For medium waves (approximately from 2 to 4 m in height), the typical WBL height is a few meters, while for larger waves (more than approximately 4 m in height), it can extend up to say 20 m. The WBL interacts with the wavy air-sea interface below and merges with the Monin-Obukhov stratified boundary layer above.

Within the WBL and according to Eq. (1), the momentum flux, to the leading order (compared to Eq. 2), can be alternatively written:

$$\boldsymbol{\tau}(z) = \boldsymbol{\tau}_\nu + \boldsymbol{\tau}_f(z), \tag{13}$$

where $\boldsymbol{\tau}_\nu$ and $\boldsymbol{\tau}_f(z)$ are the viscous stress and form stress at the height of $z$ respectively (Donelan et al., 2012). Assuming that waves' impacts decay exponentially in the vertical, the wave form stress at $z$ is defined by

$$\boldsymbol{\tau}_f(z) = \rho_w \int\limits_{k_{min}}^{k_{max}} \int\limits_{-\pi}^{\pi} e^{-2kz} \beta_g(k,\theta) \omega F(k,\theta) \boldsymbol{k} d\theta dk, \tag{14}$$

where $F(k,\theta)$ is the 2D wave variance spectrum, $\theta$ denotes the wave direction, $\rho_w$ is the water density, and $k_{min}$ and $k_{max}$ are the minimum and maximum wavenumbers. $\beta_g$ is the wave growth rate as a function of wind speed at the height of $\lambda/2$ (i.e. $u_{\lambda/2}$ calculated by the logarithmic wind profile, where $\lambda$ is the wavelength):

$$\beta_g(k,\theta) = A\omega \frac{\rho_a}{\rho_w} \frac{[u_{\lambda/2}\cos(\theta-\theta_w)]|u_{\lambda/2}\cos(\theta-\theta_w)|}{c^2}, \tag{15}$$

where $\theta_w$ indicates the wind direction, $c$ is the wave phase speed, and the proportionality coefficient $A$ is expressed by (Donelan et al., 2012):

$$A = \begin{cases} 0.11 & u_{\lambda/2}\cos(\theta) > c & \text{(wind sea)} \\ 0.01 & 0 < u_{\lambda/2}\cos(\theta) < c & \text{(fast running swell)}. \\ 0.1 & \cos(\theta) < 0 & \text{(swell opposing the wind)}. \end{cases}$$

The form drag is calculated using the friction velocity ($u_* = \sqrt{\rho_a^{-1}|\boldsymbol{\tau}'|}$) and the wind speed at a reference height of $z$ (i.e. $u_z$) as $Cd_f = (u_*/u_z)^2$. The viscous stress is expressed by

$$\boldsymbol{\tau}_\nu = \rho_a Cd_\nu' |u_z| u_z, \tag{16}$$

where the adjusted viscous drag by the form drag (sheltering effect) is given by

$$Cd_\nu' = \frac{Cd_\nu}{3} \left(1 + \frac{2Cd_\nu}{Cd_\nu + Cd_f}\right), \tag{17}$$

where $Cd_\nu$ is the viscous drag coefficient.

## 3 Data

### 3.1 Dataset

The FINO1 measurement mast in the North Sea is located about $45$ km north of the Borkum island, Germany. Its geographical coordinates are $54°0'53.5''$ N, and $6°35'15.5''$ E. The water depth at FINO1 is approximately $30$ m and the mast height is $100$

m above the mean sea level. The site is exposed to an unlimited fetch area for northwesterly and northerly winds (Bakhoday-Paskyabi et al., 2018). The mast is equipped with different meteorological sensors such as cup anemometers to measure the velocity at 33, 40, 50, 60, 70, 80, 90 and 100 m, and sonic anemometers with a sampling frequency of 10 Hz at 40, 60 and 80 m (Fig. 2b). During the OBLEX-F1 campaign between 2015 and 2016 two additional Gill R3-100 sonic anemometers were installed at 15 and 20 m above the mean sea level with sampling frequency of 25 Hz. The orientation of sonic anemometers was set at 135 degrees, which means that the wind shadow zone extended approximately above 245 degrees. While this paper's approach is primarily based on sonic anemometer wind data at 15 m height, I also make use of the 20 m sonic wind velocity data to improve the evaluation of the proposed theoretical coherence model in this study. This is crucial for a better understanding of the developed model's performance and applicability. Additionally, considering the significance of coherence function representation in relation to offshore wind turbine blade loads, this information is quite insightful for not large vertical separation distances. Furthermore, I use wave frequency spectra recorded by an AXYS wave buoy in the close vicinity of FINO1 platform during the study period.

## 3.2 Data Analysis

Figure 2 shows time series of wind, wave, and stability parameter during the study period in June 2015. We do not exclude the effect of flow distortions by the FINO1 mast when the spectra show very clear wave-induced elevation (for wind directions between 245° and 360°), see one of quality criteria in Appendix B. Sonic data at 15 m height scaled to 10 m height, i.e. $U_{10}$, using MOST shows a range of wind speeds from moderate to high. ($2 \leq U_{10} < 11 \text{ ms}^{-1}$, Fig. 2a), and during the strong wind-wave interactions (after June 24), the wind and wave directions are mostly misaligned (with direction differences larger than 100°). I study few cases, specifically two cases representing the opposed-wind and swell conditions (vertical dashed lines). To investigate coherence, I analyze two additional dates for which we have concurrent measurements of high-frequency wind at heights of 15 m and 20 m, both displaying clear wave peaks in their spectra, see Figs 6a and b.

Figure 2c shows the atmospheric stability parameter $z/L$ at a height of $z = 15$ m where

$$L = -\frac{u_*^3 \bar{\theta}_v}{g\kappa(\overline{w'\theta_v'})}$$

denotes the Obukhov length scale (in meter), $\kappa$ and $g$ are the von Kármán constant and the gravitational acceleration, respectively. $\overline{w'\theta_v'}$ is the flux of virtual potential temperature, and $\bar{\theta}_v$ denotes the virtual potential temperature. The stability changes from stable ($L > 0$) to unstable ($L < 0$), and both coherence and decomposition study episodes represent, on average, stable conditions.

## 4 Results

In this section, I utilize three-dimensional wind measurement for calculating measured turbulence and wave-induced stresses. For clarity and brevity, the vertical wind component is used to study the performance of the wave-turbulence decomposition techniques. This further provides insights into vertical motion relevant for studying vertical momentum transport, vertical

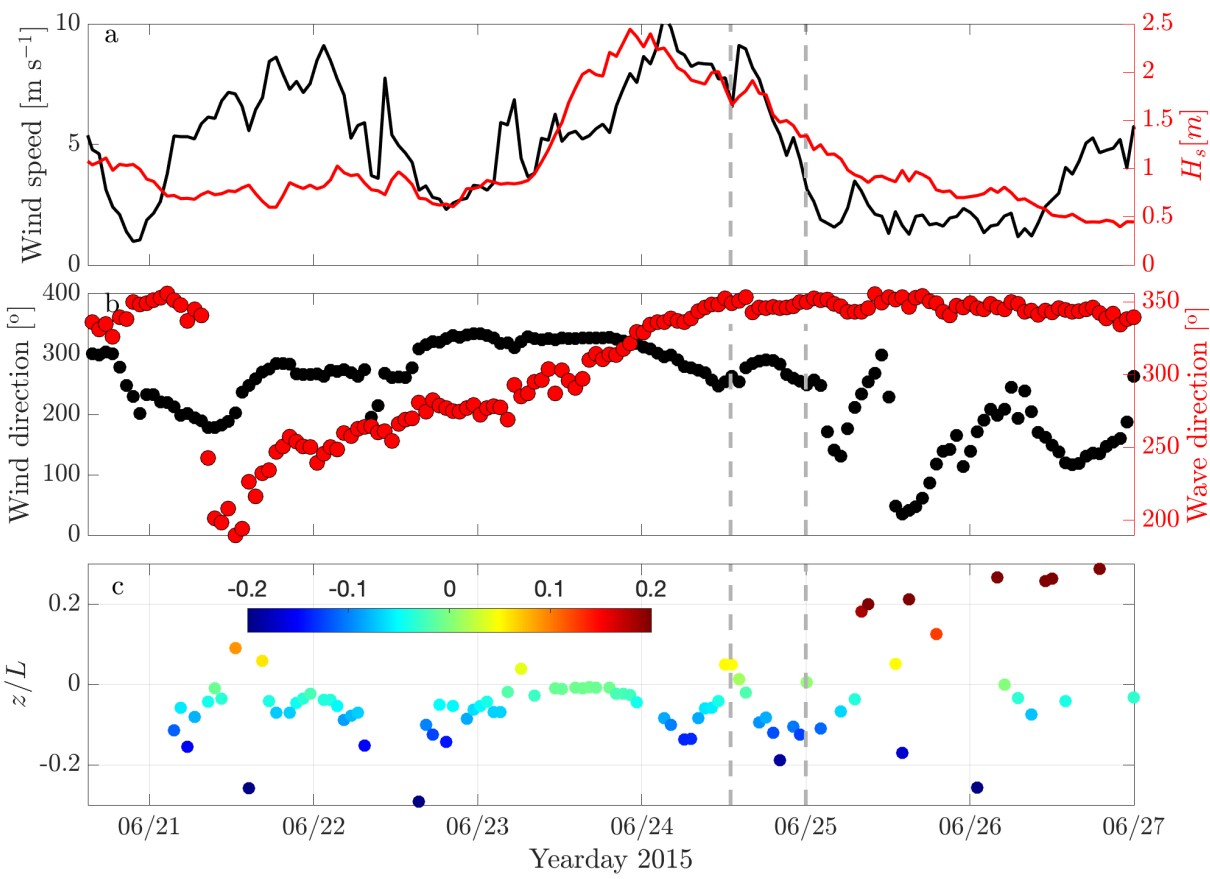

**Figure 2.** (a) Times series of wind speed at 10 m height measured by the FINO1's cup anemometer (black line) and significant wave height (red line) measured by a floating buoy operating in the close vicinity of FINO1 platform; (b) wind (black markers) and wave (red markers); and (c) the values of the stability parameter from which the Obukhov length, $L$, was calculated from sonic measurement at height of 15 m above the mean sea level collected with sampling frequency of 25 Hz between 21 and 27 June 2015. The stability classes have been color-coded in this figure. Vertical grey dashed lines highlight the study events of wind-wave interaction at 24[th] June at 13:00 and 25[th] June at 00:00, respectively.

coherence, and turbulence variation with height. I initiate this section with idealized examples, based on the proposed coherence model, to establish a foundational understanding of the developed techniques under controlled parametric conditions.

## 4.1  Theoretical coherence

The theoretical (normalised) coherence spectra for two different wind speeds are obtained using Eq. (8) in an ideal setup. Figure 3a shows that the coherence functions have peak values at $f_p$ with an exponential decay beyond the wave band, where $f_p$ is the wave peak frequency in Hz. The decay coefficients for both wind and wave components in Eq. (8) are set to constant values for the sake of simplicity ($C_1 = C_2 = 1$) and do not increase by increasing the wind speed. From the procedure given in Section 2.2 and coherence information drawn in Fig. 3a, I obtain realisations of turbulent winds for two cases, with wind speeds of $U_{10} = 10$ and 20 m/s respectively. For simplicity, the wind sea wave bulk parameters are treated as total bulk parameters (i.e. $H_s$ and $T_p$). The spectral energy distributions of synthetically generated winds are shown in Fig. 3b that identifies the impact of waves on the turbulence in the inertial subrange, particularly across the wave frequency band, $0.6f_p \leq f \leq f_p + 0.1$. Figures 3c and d depict the fitting of the model spectrum derived from Eq. (3) for vertical wind. This process is carried out to eliminate the wave peak and to synthetically generate the spectra of turbulent fluctuations within the wave frequency band (indicated by the blue curves, as discussed in Section 2.1). The parameters presented in this figure are idealized to enhance conceptual clarity and technical demonstration, rather than reflecting real-world values. They serve as a theoretical framework to facilitate detailed illustrations of the underlying concept and are not representative of practical measurements or actual conditions. It is important to note that the actual values and more detailed information for these parameters, obtained through fitting to the observed coherence data, can be found in Appendix C and Table C1, as well as Fig. 6a and b.

## 4.2  Measured wind-wave spectra

Figure 4a shows the time variation of the wave age, $\chi$, during the study period covering both the mixed wind-sea (i.e. $\chi < 1.2$) and the swell waves (i.e. $\chi \geq 1.2$). Wind and waves are obviously aligned during the wind-sea conditions. For the conditions where the swell waves are dominant (see Fig. 4b), wind and waves are mainly misaligned with a difference approximately more than $100°$. This is particularly the case after 25 June when the wind-wave misalignment shows an oscillation-like behaviour. The atmosphere experiences mainly stable and also in lesser extent unstable conditions during this period (see Fig. 2c). The time-evolution of the power spectral density of the vertical wind speed (i.e. $w$-component) is presented in Fig. 4c. This figure has been overlaid by the time series of the wave peak frequency, $f_p$. There is a good agreement between the measured $f_p$ and the spectral peak of the measure vertical component wind at the wave frequency band, consistent with an increase in the values of wave age (i.e. values of $\chi$ greater than 1.2). The agreement is more pronounced under stable atmospheric conditions, with somewhat weaker agreement observed under unstable conditions, especially after June 25.

To further investigate the ability of the suggested method in splitting the fluctuations of wave and turbulence, I use 30-min time series of sonic anemometer data at 15 m height for two study events, depicted in Fig. 5. These events correspond to strong swell-wind interaction, characterised by low wind, large values of $\chi$, and spectrally enhanced energy within the wave-affected frequency band (i.e. around $f_p$). The spectra of the corrected vertical velocity fluctuations $w'$ and the vertical wave orbital

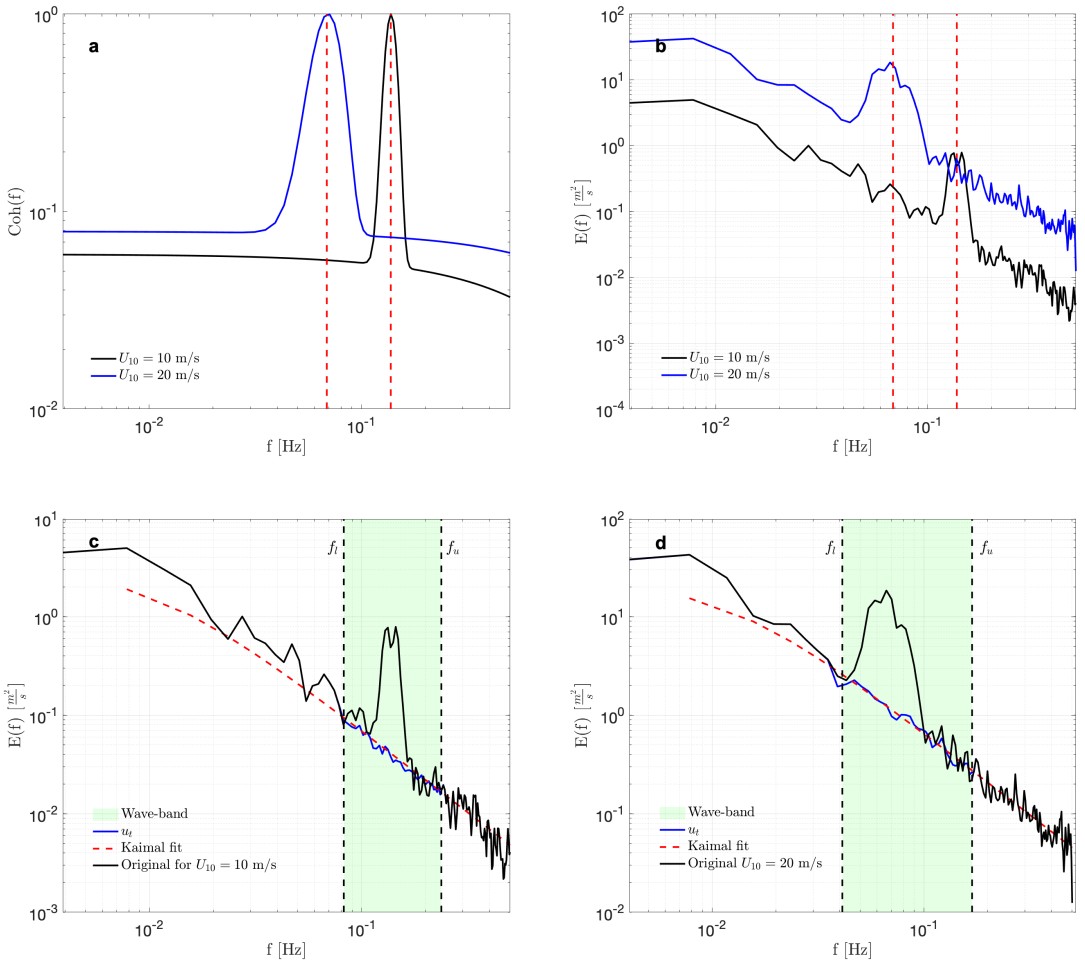

**Figure 3.** (a) Coherence functions of the wind and wave for $A_1/A_2 = 7.5$ in Eq. (8) (before normalising the total coherence and only Term2) with peak wave frequencies of $f_p = 0.14$ Hz and $f_p = 0.06$ Hz corresponding to wind speeds of 10 and 20 m/s respectively; (b) the energy spectra of the vertical velocity $w$ estimated from the Kaimal spectra Eq. (3) for two study cases, as shown by vertical dashed lines in Fig. 2; (c) the energy spectrum of the first case with $U_{10} = 10$ m/s and $f_p = 0.14$ Hz (black line) and the corrected spectrum (blue line); and (d) the energy spectrum of the first case with $U_{10} = 20$ m/s and $f_p = 0.06$ Hz (black line) and the corrected spectrum (blue line). The red dashed lines drawn in (c) and (d) show the spectral curves calculated from Eq. (3). Furthermore, the green-coloured areas in these figures represent the wave-affected frequency band with lower and upper frequencies of $f_l = 0.6 f_p$ in Hz and $f_u = f_p + 0.1$ in Hz, respectively.

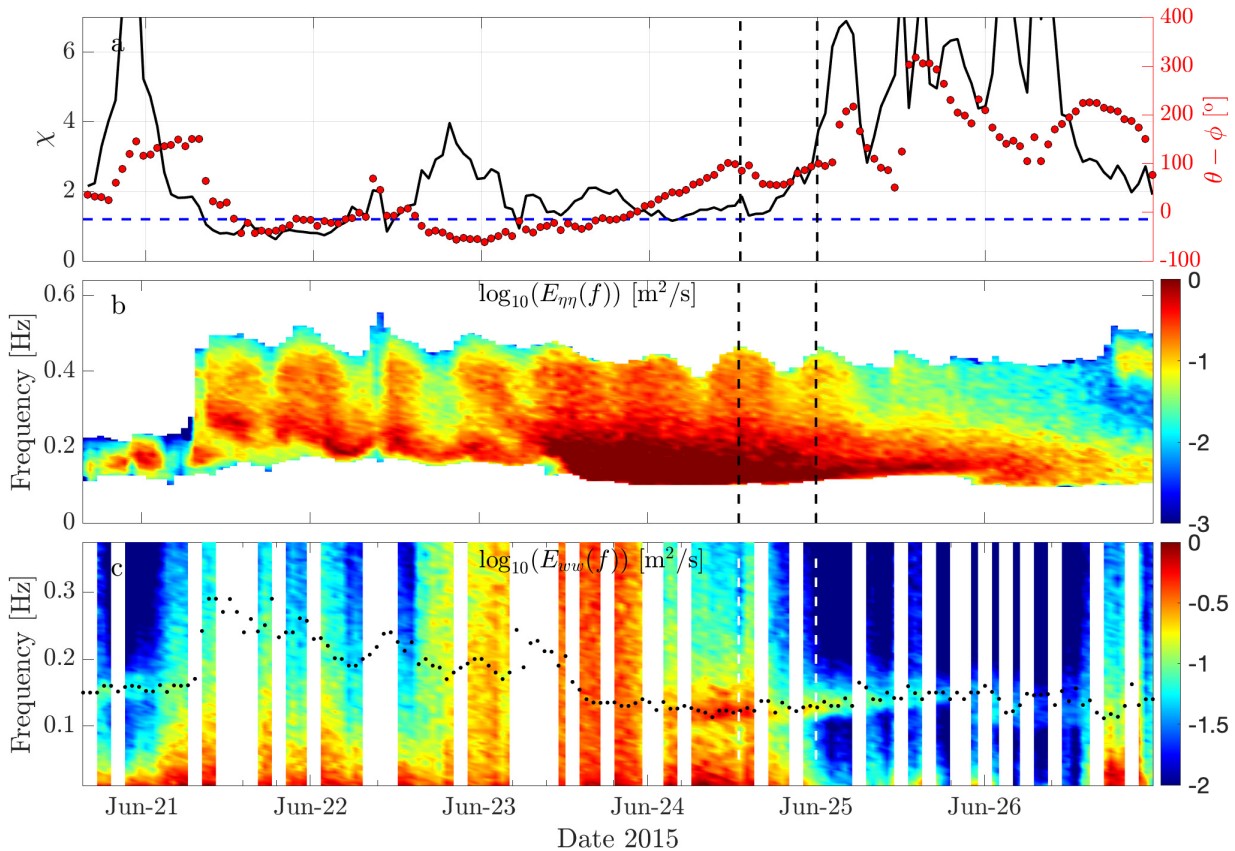

**Figure 4.** Time series of wave age $\chi = c_p/U_{10}$ in which $U_{10}$ denotes the wind speed at 10 m height and $c_p$ is the phase speed. This figure contains the wind-wave misalignment timeseries $\theta - \theta_{wnd}$ where $\theta$ and $\theta_{wnd}$ denote the wave and wind directions respectively. The blue-dashed line ($\chi = 1.2$) represents the separation limit between the wind see and swell (i.e. $\chi \geq 1.2$); (b) spectral energy evolution of surface wave elevation $\eta$ measured from AXYS buoy operating in very close vicinity of FINO1 met-mast; and (c) time evolution of energy spectra for the wind $w$-component in June 2015 between 21 and 27 calculated from the 15 m height sonic anemometer with a sampling frequency of 25 Hz. The black dotted markers are the wave peak frequencies calculated from the buoy measured peak wave period $T_p$. Vertical grey dashed lines indicate the study cases.

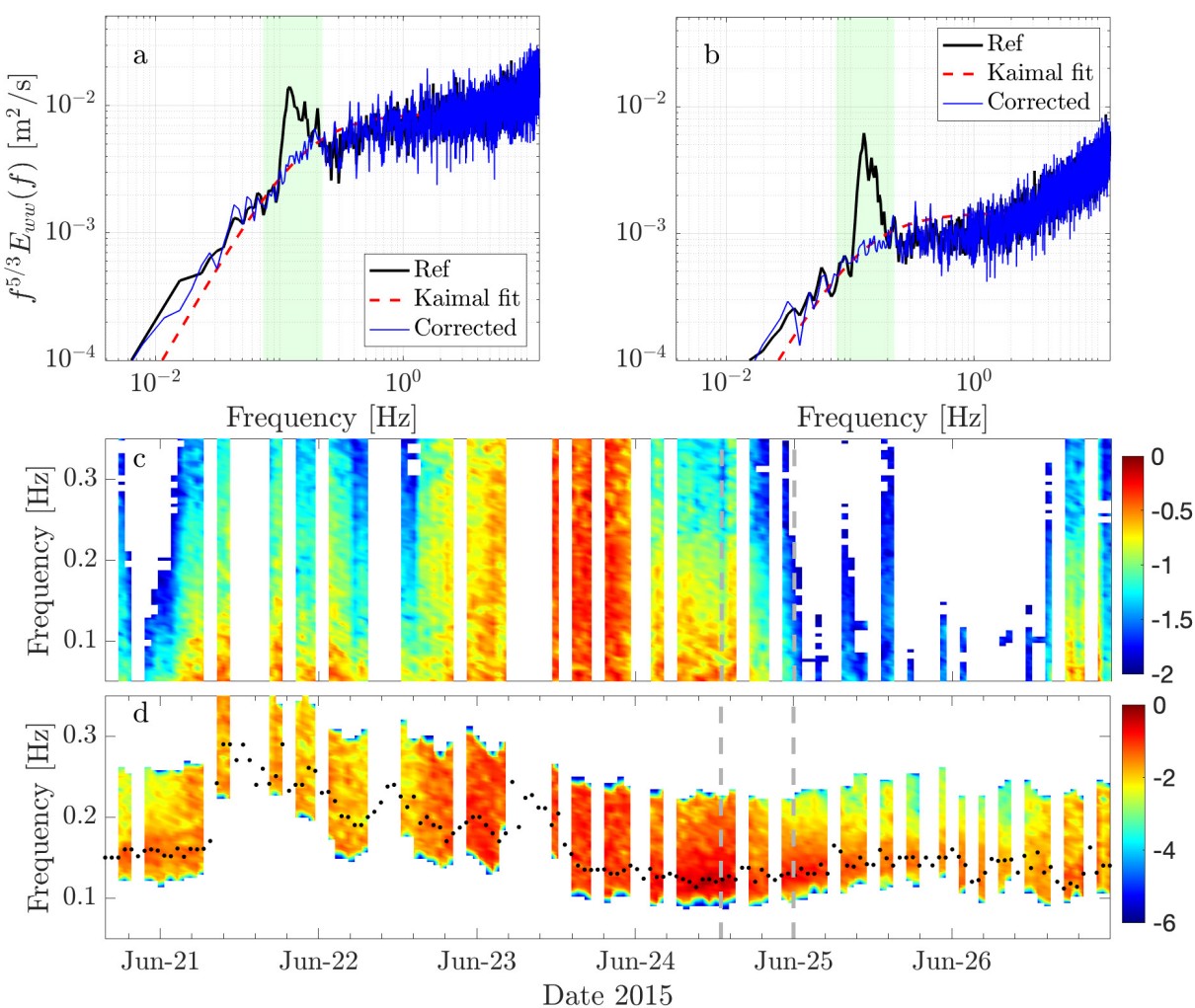

**Figure 5.** (a,b) Scaled power spectra of vertical velocity fluctuations $w'$ for two events drawn as vertical grey dashed lines in Fig. 2a, and the corresponding corrected (decomposed) spectra (blue curves)). The red dashed lines show the spectral curves calculated from Eq. (3); and (c,d) energy-time spectra of decomposed vertical velocity $w'$ and the wave-induced vertical velocity $\tilde{w}$ fluctuations by the use of suggested decomposition algorithm. The black dotted markers in (d) are the wave peak frequencies extracted from buoy measurements.

velocity $\tilde{w}$ are shown in Fig. 5c and d. It is observed how the decomposition detaches the wave-induced energy elevation from the inertial subrange of $w$. To assess the effectiveness and performance of the proposed decomposition technique, we compare the power spectra of the vertical velocity for three decomposition methods that depend exclusively on the time series of high-frequency wind at a single height. This comparison is presented in Figure D1 (see Appendix D).

## 4.3 Spectral and coherence analyses

Figure 6 displays the coherence spectra of 15-minute vertical wind data at heights of 15 m and 20 m, with a separation distance of 5 m. The two cases depicted in this figure were measured under stable atmospheric conditions and low wind. During the first period, the significant wave height was approximately 0.8 m, and the peak wave period was around 7 s (Figs 6a and c). Consequently, this 15-minute segment encompassed roughly 128 swell periods, providing a sufficiently robust estimate of coherence. In this scenario, we can clearly observe the presence of a wave-induced elevation. Figure 6c shows the power spectrum of vertical velocity fluctuations. Initially, the spectrum is primarily influenced by surface waves at frequencies within $0.6 f_p - f_p + 0.1$ Hz, corresponding to a wave peak period of 7 s, with a notable peak in this range. At higher frequencies, the spectrum closely followed the Kolmogorov $-5/3$ curve up to 5 Hz (not shown). Using the coherence-based method described in Eq. (9), It is possible to separate the total velocity fluctuations into wave and turbulent components, as depicted by black curves. In Figs 6a and b, it is evident that scales at frequencies greater than $f_p + 0.1$ Hz exhibit a sustained coherence value of approximately 0.1. This could be attributed to data characteristics and quality, or potential effects of residual flow distortion during low-wind conditions.

Figure 7a displays time series of original (red curve), corrected (black curve), and wave-induced (blue curve) vertical velocity during a segment on June 26 at 04 (refer to Fig. 6b). The black and blue curves illustrate the corrected and estimated wave time series, respectively, extracted directly using the decomposition method developed in this paper (see Sec. 2.1). To examine the impact of corrections on turbulence statistics, the effects of corrections on the coherence spectrum and structure function are illustrated in Figs. 7b and c respectively. Utilizing the decomposition method at heights of 15 m and 20 m to compute the corrected coherence results in a significant reduction in coherence within the wave-affected frequency band, although some small wave-correlated points still remain in this frequency range (i.e. green area in Fig. 7b). Furthermore, we observe a more significant decline in coherence at frequencies above the wave band.

Starting from the time series of vertical wind velocities $w(t)$, I define the second-order structure function as follows:

$$S^2(t) = \left\langle |w(t + \tau_{inc}) - w(t)|^2 \right\rangle, \tag{18}$$

where $\tau_{inc}$ indicates a time lag, and $< \cdot >$ denotes an average over all time lags. By having knowledge of $S^2(t)$, it is possible to transform this into a function in space $S^2(r)$ using the parameter $r = \tau_{inc}\bar{u}$, where $\bar{u}$ represents the mean wind speed averaged over the entire time series, based on Taylor's hypothesis. The values of $r$ vary between the Taylor length scale and the integral length scale. In Fig. 7c, the non-corrected structure function exhibits wave-induced oscillations that gradually dampen at larger spatial scales. The shape of $S^2(r)$ indicates that wave orbital contamination influences the slope of the second-order structure function for ranges before the onset of the oscillating tail. The application of decomposition to generate corrected time series in computing the structure function eliminates not only these wave-induced oscillations but also the slope-enhancement from the second-order structure function in this figure.

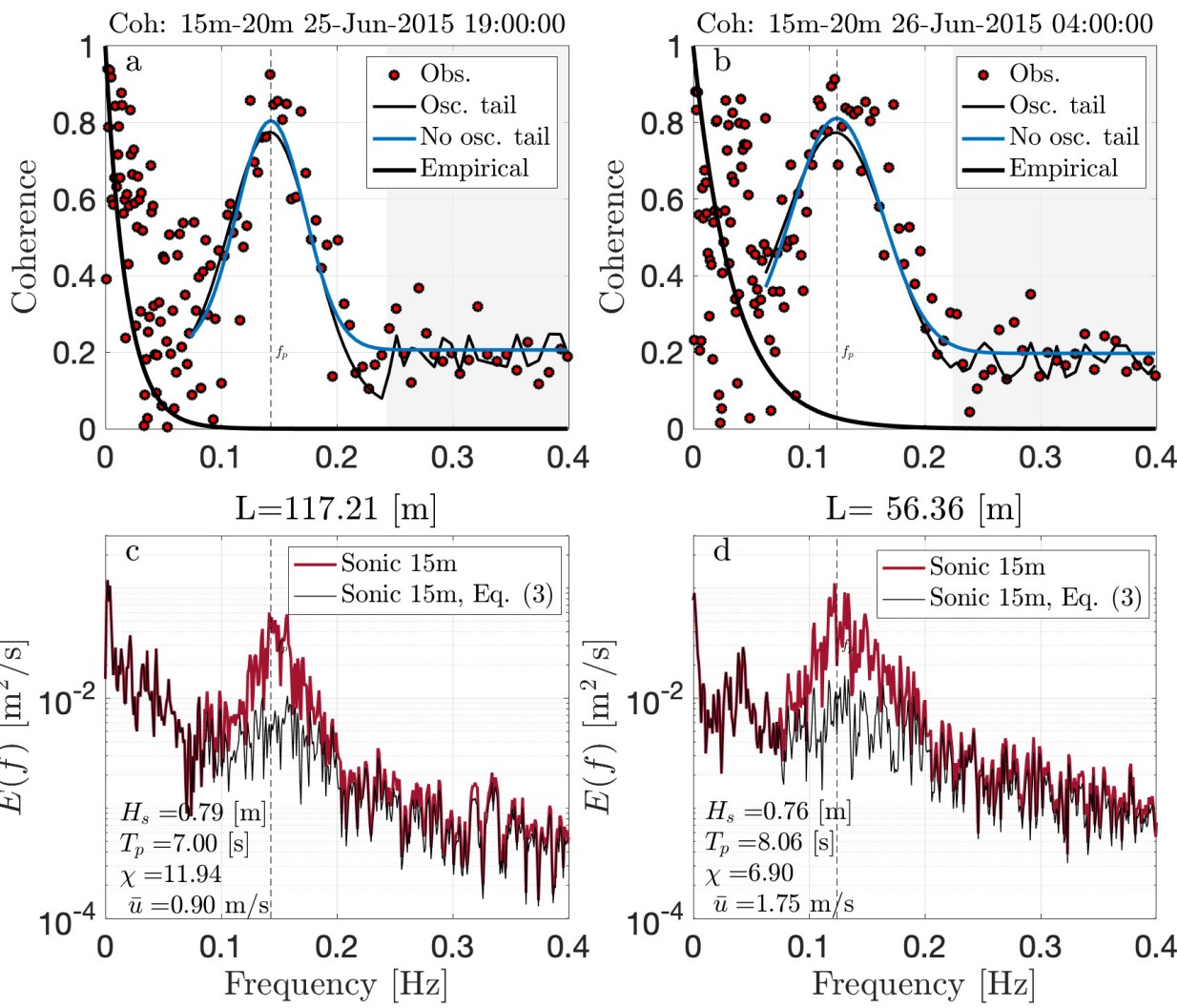

**Figure 6.** (a,b) The coherence spectra of vertical velocity time series at two heights (15 m and 20 m) are marked in red. In these plots, the thick black curves represent the empirical/theoretical coherence function, while the black thin and blue curves represent the fitted coherence derived from Eq. (8). The black thin curves illustrate the impact of wave-induced oscillations in the coherence tail (shaded regions); and (c,d) power spectra of the original vertical velocity fluctuations for two distinct dates during stable atmospheric conditions, depicted by the red curves. The black curves represent the spectra of turbulent components obtained directly through the coherence-based correction based on Eq. (9).

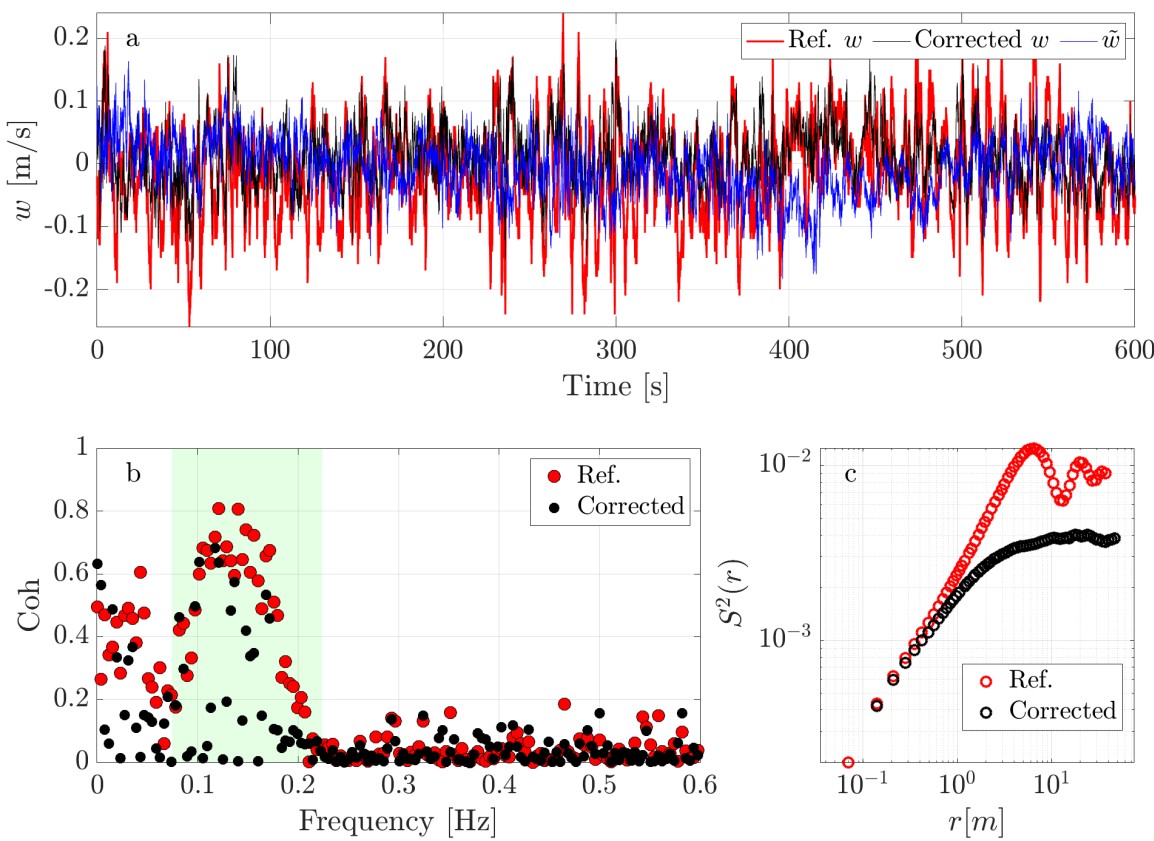

**Figure 7.** (a) Time series of the wave-contaminated vertical velocity (red curve), the corrected $w$ (black curve), and the wave component $\tilde{w}$ (blue curve) obtained by applying the developed decomposition method; (b) the contaminated and corrected coherence for a separation distance of 5 m using data from both 15 m and 20 m sonic anemometers. The coherences are computed using Eq. (8); and (c) the second-order structure function calculated from sonic anemometer measurements at 15 m height at the same time as illustrated in Fig. 6b.

## 4.4 Momentum flux estimation

Regarding to the calculation of momentum flux, splitting between wave and wind fluctuations is not reasonable if there is no obvious footprint of waves in the measured velocity spectra at 15 m height. As a result, the decomposition algorithm is applied if the ratio of the energy variances in the wave frequencies and model spectrum, $R$, is larger than 1:

$$R = \frac{\int_{f_l}^{f_u} E_{ww}^o(f)df}{\int_{f_l}^{f_u} E_{ww}^m(f)df} > 1, \tag{19}$$

where $E_{ww}^o$ and $E_{ww}^m$ are the sonic based energy spectrum and the model spectrum of the vertical wind speed given by Eq. (3), respectively. Figure 8a shows a measure to assess the strength of wavy structures in the observed velocity spectra. I apply the decomposed turbulence time series when $R > \bar{R}$ (i.e. when there exists a well-pronounced energy elevation around $f_p$). Here, $\bar{R}$ represents the average value of $R$. In Fig. 8b, I compare the total wind stress at the surface obtained from Eq. (13), black curve, with the bulk estimation (red curve) derived from COARE3.6 algorithm (see Eq. 2). The two stress estimates align consistently when $R < \bar{R}$. Additionally, it is noteworthy that the wave-induced form stress at the surface, shown by blue markers, undergoes transitions from positive to negative for swells moving opposite to the wind direction (i.e., when $R > \bar{R}$). Figure 8c shows that the estimated form stress at $z = 15$ m according to Eq. (14) is approximately in acceptable agreement with the measured $|\tilde{\tau}|$ from the sonic data, using eddy covariance technique according to Eq. (2). In estimated form stress at 15 m height, the dimensionless function for the vertical decay, i.e. Eq. (14), plays a significant role in vertical distribution of the wave-induced momentum flux. Moreover, the decomposition method outlined in Section 2.1 has been applied to all three components of measured wind velocities to estimate the observed wave-induced stress $|\tilde{\tau}|$ represented in Eq. (2). Figure 8d illustrates that the ratio of wave-induced (turbulence) intensity (the standard deviation of $\tilde{w}$ over the mean wind speed) to corrected turbulence intensity (the standard deviation of the corrected $w$ over the mean wind speed) is most pronounced when wave elevations are clearly visible around the peak frequency $f_p$ (in agreement with Fig. 8a).

## 5 Conclusions

I have suggested a wind-wave decomposition algorithm for the turbulent airflow over the ocean in the presence of swell waves based on high frequency data recorded from a sonic anemometer at 15 m height above the mean sea level. The wave-turbulence decomposition method proposed in this study possesses some key characteristics: (1) It relies solely on sonic wind velocity data, eliminating the need for simultaneous high-frequency wave measurements in the decomposition process. It assumes turbulence field stability during transformation into wavenumber space and disregards velocity fluctuations within the wave band; (2) the method uniquely adopts a statistical approach, employing a turbulence spectral model to effectively bridge the gap between high- and low-frequency sections in the observed spectra. This allows for the estimation of the variance attributed to turbulent velocity fluctuations within the wave frequency band, relying solely on the energy spectrum of the corresponding wind component; and (3) notably, this method provides wind-corrected and wave time series, a crucial data for structural analysis that, to the best of my knowledge, is not available through many known methods.

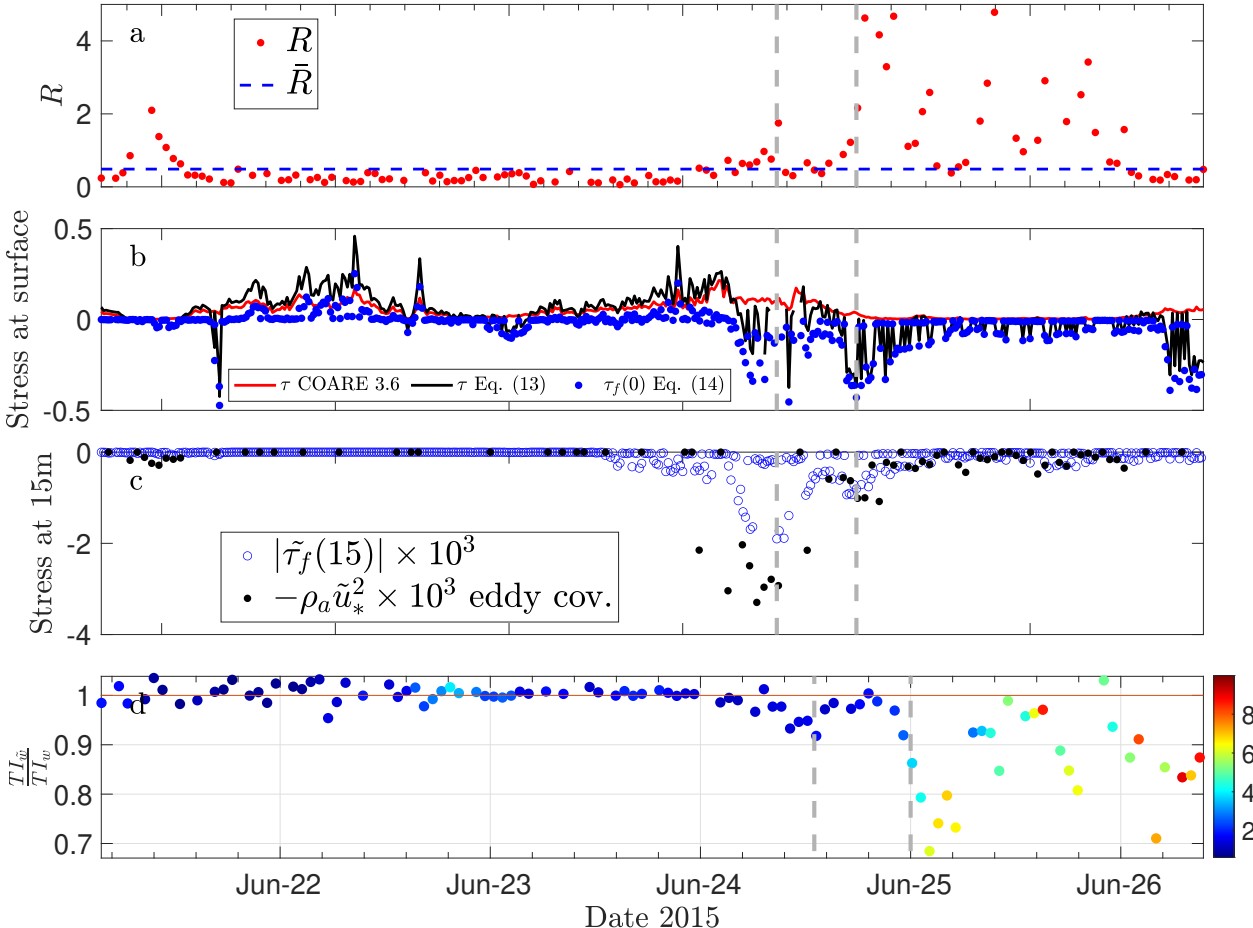

**Figure 8.** Time series of (a) the wave-turbulence strength ratio $R$; (b) momentum stresses at the surface calculated using COARE 3.6 algorithm (red line), Donelan et al. (1999) parameterization Eq. (13), black line, and the wave-induced form stress estimated from Eq. (14) at the surface; (c) the momentum fluxes at 15 m height estimated using decomposed turbulent wind data based on eddy covariance technique, black markers, and the estimated wave-induced form stress at measurement height $z = 15$ m using Eq. (14), i.e. $|\boldsymbol{\tau}_f(15)|$, blue markers. Here $\tilde{u}_* = \sqrt{\rho_a^{-1}|\tilde{\boldsymbol{\tau}}|}$; and (d) the ratio of the wave induced turbulence intensity and the corrected turbulence intensity, $TI_{\tilde{w}}/TI_w$,

Furthermore, I've introduced a theoretical formulation for coherence that considers both wind fluctuations and the influence of swell waves. This model has been utilized not only to create idealized wind time series under swell conditions but can also be adjusted using observational data. In this case, I used additionally sonic data at 20 m (along with sonic data at 15 m) to estimate observed coherence and determine the fitting coefficients presented in Eq. (8). Furthermore, I quantified the wave-induced stresses by assuming both a certain growth rate and a vertical decay function. Few real cases during the opposed wind-wave conditions were selected to demonstrate physical aspects of wind and waves regimes. Under the wind-sea conditions, it was found that the wind stresses were almost similar to those observed under atmospheric neutral conditions. This is because the height of wave boundary layer was below the height of measurement, and waves could contribute to a very small fraction of the total wind stress.

Although I have introduced a theoretical coherence function with a few number of fitting parameters, there is potential for improving the model and its efficiency by reducing the number of adjustable parameters and the model mathematical representation. Further details on this, as well as the reconstruction of the turbulence box for assessing the structural loads of offshore wind turbines over swell waves, are discussed in a separate independent study.

*Code availability.* Some codes to plot figures in this paper will be available on a valid access request.

*Data availability.* Time series of wind, wave age, and stability, covering yeardays between 160 and 210 2015, have been made available on https://doi.org/10.5281/zenodo.7422388. The OBLEX-F1 high frequency data (used in this study) at 15-m measured by sonic anemometer are on https://doi.org/10.5281/zenodo.7591198 and can be alternatively available through a valid access request.

## Appendix A: Calculation of $df/dk$ based on linear dispersion relation

The wavenumber and frequency spectra are interrelated through the dispersion relation ( I assume here the linear dispersion relation):

$$\omega^2 = gk\tanh(kd),$$

where $\omega$ is the angular frequency, and $d$ denotes the water depth. The spectral variance, whether expressed in frequency or wavenumber spectra, can be determined accordingly:

$$\sigma^2 = \int F(k)dk = \int E(f)df,$$

where $E(f)$ and $F(k)$ are the frequency and wavenumber spectra respectively. Assuming linear dispersion, we can estimate $dk/df$, essential for the transformation between these two spectra as

$$\frac{df}{dk} = \frac{g}{4\pi\omega}\left[\tanh(kd) + \operatorname{sech}^2(kd)kd\right].$$

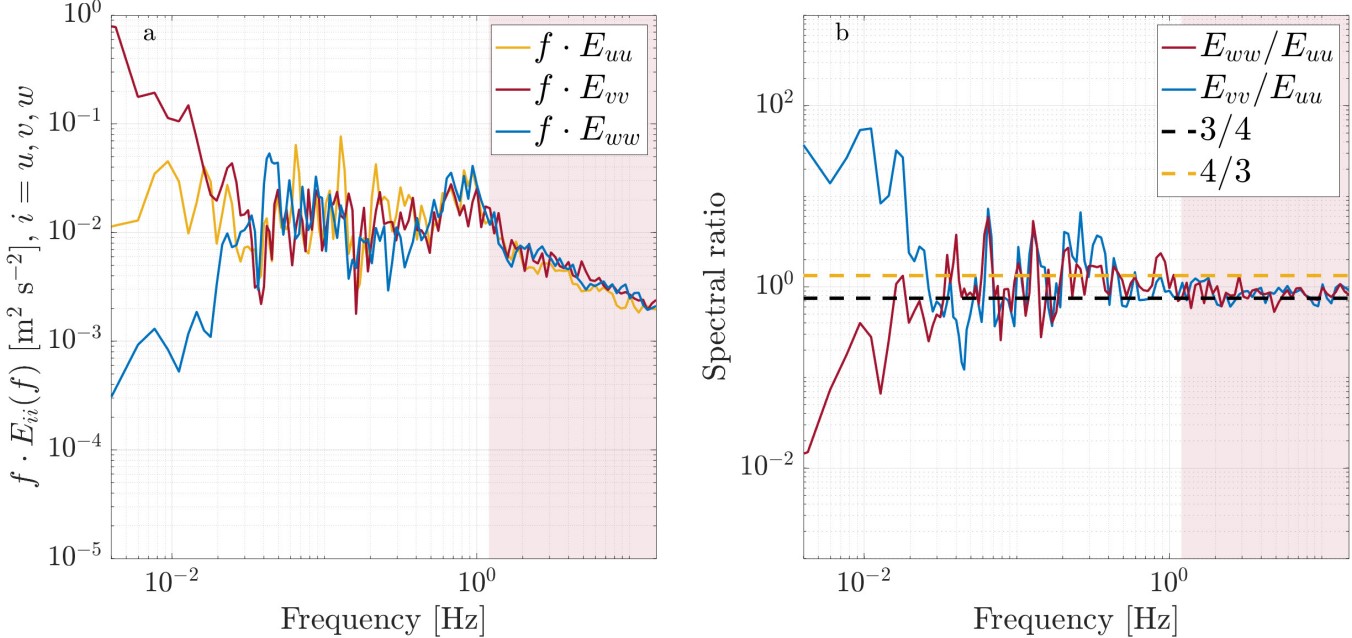

**Figure B1.** (a) Three-dimensional velocity spectra $f \cdot E_{ii}(f)$ for $i = u, v$, and $w$ showing the spectral regions of the wave-affected peaks and the inertial subrange; and (b) velocity spectra ratios with horizontal dashed yellow and black lines indicating the value of $4/3$ and $3/4$, respectively.

**Appendix B: Isotropic ratio**

Here, I examine one of the quality criteria, which is the presence of the isotropic portion of the spectrum in the inertial subrange (outside the wave-contaminated frequency range). The cross-coherences, at zero separation, are not computed to assess isotropy, as not all of these cross-coherences may be exactly zero. To be more specific, I define the isotropy criteria as follows:

$$\frac{E_{u'u'}}{E_{w'w'}} \sim \frac{E_{u'u'}}{E_{v'v'}} \sim \frac{3}{4}.$$

It is worth noting that these ratios over the isotropic bandwidth converge to 3/4 (Fig. B1 red-colored areas). In Fig. B1, the power spectra and spectral ratio of the non-corrected time series are presented to assess the statistical isotropy of the three velocity components. It is evident that the non-corrected ratios in Fig. B1b approach a value of 3/4 for frequencies larger than wave-affected band. In some cases with wave peaks in the velocity spectrum during low wind conditions, I notice increased anisotropic features (not shown), likely due to higher wave contamination beyond the wave-frequency band
(Bakhoday-Paskyabi, 2019) and partial effects of flow distortion.

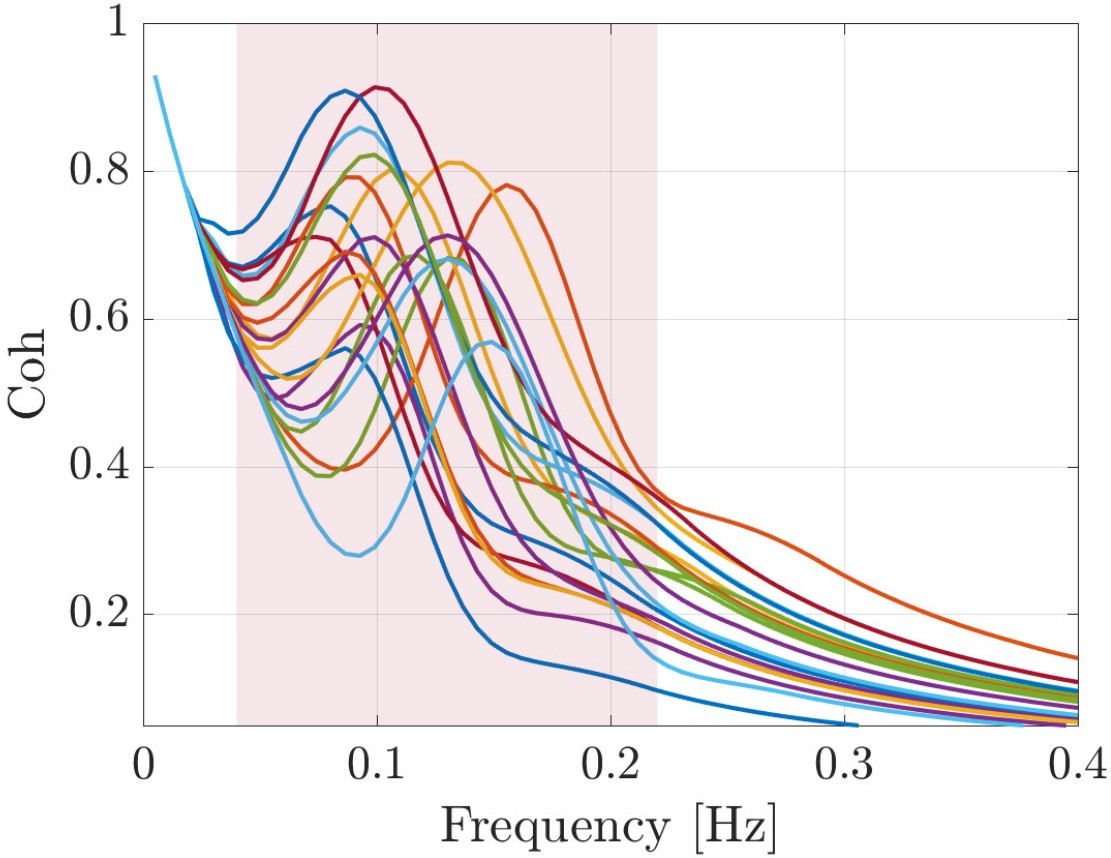

**Figure C1.** Theoretical coherences are generated based on the fundamental parameters listed in Table C1. To create 20 ensembles, I perturb $H_s$ within the range of $[0.5, 1.5]$, $T_p$ within $[5, 12]$, $A_1$ within $[0.1, .5]$, and $A_2$ within $[0.02, 0.1]$. It is important to note that in the theoretical model defined in Eq. (8), we only use Term1 and Term2 to produce this figure.

**Appendix C: Theoritical coherence function in Eq. (8)**

To obtain estimations for the fitting parameters in the theoretical coherence model outlined in Eq. (8), I provide the results of the fitting process in Table C1, for the scenario illustrated in Fig. 6b. The fitting model comprises only Term1 and Term2. To generate multiple ensembles using the derived parameters, I perturbed four critical parameters, including $H_s$, $T_p$, $A_1$, and $A_2$.

**Appendix D: Wind-wave decomposition methods**

The choice of decomposition method in this manuscript is based on specific considerations related to the research objectives and the nature of the data. The approach solely utilizes sonic wind velocity data at a single height, omitting the need for concurrent

**Table C1.** Theoretical model parameters are determined by applying the least squares estimation method to the observed coherence illustrated in Fig. 6b. The total wave bulk parameters used to generate this figure are $H_s = H_s^{sw} + H_s^{ws}$ and $T_p = T_p^{sw} + T_p^{ws}$ where $H_s^{sw}$ [m] and $T_p^{sw}$ are the significant wave height and peak period of swell waves, and $H_s^{ws}$ and $T_p^{ws}$ are the wind-sea bulk parameters estimated using Carter's formula, Sec. 2.2.

| $A_1$ | $A_2$ | $C_1$ | $C_2$ | $\beta_2$ | $\bar{u}$ [m/s] | $H_s^{sw}$ [m] | $T_p^{sw}$ [s] |
|---|---|---|---|---|---|---|---|
| 0.3962 | 0.5434 | 0.3 | 0.07985 | 0.07443 | 1.7 | 0.7 | 8 |

high-frequency wave measurements in the decomposition process. In this appendix, I compare three existing wave-turbulence decomposition methods that rely only on the high-frequency measurement of wind at a single height. These methods are: the stopband (SB) method, spectral linear transformation, and an Empirical Mode Decomposition (EMD) based approach. The first two are established filtering techniques, while the latter is a novel signal processing method.

The stopband filter eliminates frequency bands in which waves dominate in the sonic velocity time series (blue curve in Fig. D1b). In this method, we employ a second-order Butterworth filter. The stopband frequency thresholds, which consist of lower and upper cutoff frequencies, are specifically designed to attenuate frequencies within the wave-affected band. The spectrum produced by the stopband filter method exhibits a significant drop in energy at wave frequencies, resulting in an underestimation of turbulent energy. The second method, adapted from Rieder and Smith (1998), involves identifying the wave frequency band in the velocity spectrum. Subsequently, we remove the wave-correlated portion of the spectrum and replace the removed frequencies with a line, as illustrated in Figure D1b (red line). While the effectiveness of these two methods relies heavily on the accurate choice of the wave frequency band, the line-fit method successfully eliminates orbital velocities from the analyzed signal.

The third approach, known as the EMD method, decomposes the observed signal in the time domain into multiple Intrinsic Mode Functions (IMFs). Each IMF, as a stationary stochastic process, characterizes a narrowband frequency-amplitude modulation typically associated with a specific physical process. I have adapted this technique from Qiao et al. (2016), who utilized the EMD method to extract wave signals from field velocity observations. In this method, the vertical velocity fluctuations, which include wave orbital velocities, are decomposed into $n$ IMFs and a residual fluctuation $w'_{\text{residual}}$ as follows:

$$w' = w'_{\text{IMF1}} + w'_{\text{IMF2}} + \cdots + w'_{\text{IMFn}} + w'_{\text{residual}}. \tag{D1}$$

The IMFs can be categorized into two groups: wave-correlated components and non-wave-correlated components. In the vertical velocity time series, the peak frequencies of IMF4 and IMF5 in Fig. D1a fall within the wave frequency range. Thus, we identify specifically IMF4 as the wave component that is excluded from the sum in Eq. (D1) to remove significantly wave contamination. IMF6 − 10, on the other hand, represent high-frequency signals like turbulence.

*Author contributions.* The author proposed, implemented the methods and all signal processing, and wrote the manuscript.

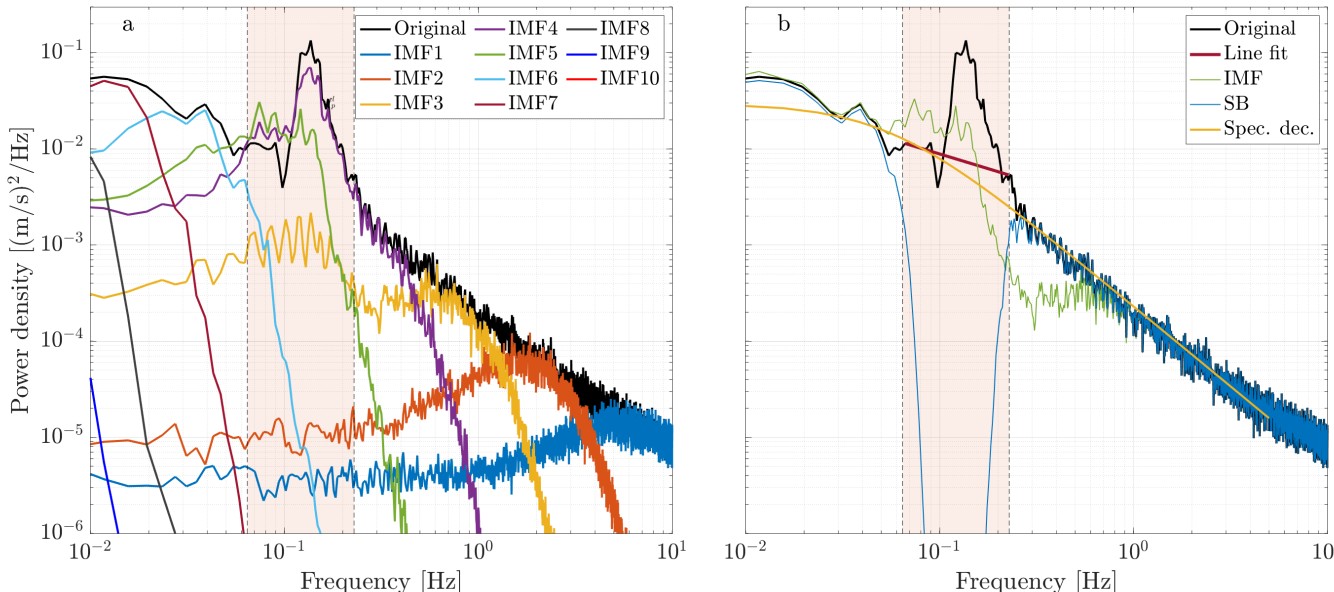

**Figure D1.** (a) Power spectra of the vertical velocity fluctuation $w'$ (illustrated in Fig. 6d) and its decomposed IMFs (for $n = 10$); and (b) comparison of different wave-turbulence decomposition methods: stopband technique (blue curve), spectral linear transformation (red line), the EMD method after removing IMF4 (green curve), and the spectral decomposition method proposed in this paper (yellow curve). The colored areas in this figure show the selected wave frequency band.

*Competing interests.* The author declares that he has no competing of interest.

*Acknowledgements.* The work is a part of the HIghly advanced Probabilistic design and Enhanced Reliability methods for the high-value, cost-efficient offshore WIND (HIPERWIND) project, which has received funding from the European Union's Horizon 2020 Research and Innovation Programme under Grant Agreement No. 101006689. Part of work supported from the academic agreement between Equinor AS and University of Bergen (through LESWIND with project number of 102239114). The simulations were performed on resources provided by UNINETT Sigma2 - the National Infrastructure for High Performance Computing and Data Storage in Norway (NN9871K and NS9696K). LiDAR data used in this study were gathered as part of the OBLEX-F1 field campaign that has been performed under the Norwegian Centre for Offshore Wind Energy (NORCOWE). The FINO1 meteorological reference data were provided by Deutsches Windenergi Institute (DEWI). Finally, we acknowledge the AXYS technologies for providing the wave frequency spectrum during OBLEX-F1 campaign.

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
