# Peer review of "Impact of swell waves on atmospheric surface turbulence: Wave-turbulence decomposition methods"

_Wind Energy Science, 2023_

## Author Comment (AC1)

Reply to reviewer: for "**Brief communication: Impact of swell waves on atmospheric surface turbulence: A wave-turbulence decomposition method**"

Reviewer comments are presented in black text using the "Calibri" font format with a size of 12. My responses are displayed in blue text using the "Calibri" font format with a size of 13.

**Reviewer 1**

Thanks a lot for your manuscript. I found it quite interesting, and I would like to suggest that this should not be reviewed as a "brief communication" but as a "research article". I am not an editor of the journal, so I do not know the formal distinctions, but I think the manuscript has sufficient material to become a paper and it is not brief (its current version has 13 pages). It actually has the size of the papers I like to read.

Thank you for your thoughtful review and valuable suggestions. I agree with your suggestion to consider the manuscript as a 'research article.' I've already discussed this with the manuscript's editor. Moving forward, I plan to expand the paper slightly by carefully addressing the specific comments provided by the reviewers. Your comments are greatly appreciated as they have already helped refine the quality and presentation of my work.

**Other main comments**

1. As I said the manuscript is quite interesting but right now it is difficult to read/follow because there is a part (I think) in which the method is applied in simulated wind fields and another part in which the method is applied to observations (I think). So it is not clear if the wind field simulations are actually used within the analysis of the observations or not.

   Thanks for your valuable feedback. It's important to note that the original format of the manuscript was a letter, which contributed to its concise nature, resulting in some limitations in providing comprehensive information and structural clarity. I understand and appreciate your concern regarding the unclear explanation. To address this, I revise and reorganize the manuscript to ensure a smoother flow and better delineation between the sections where the method is applied to

simulated wind fields and where it is applied to observations. As an example, I modify the methodology section by renaming "2.2 Synthetic turbulence" to "Wind-wave interaction: Coherence and synthetic turbulence" and positioning it as section 2.1. This adjustment, coupled with improved explanations in the methodology section, enhances the overall coherence between wind field simulation and observations, making clear their significance and interconnection.

2. This is not clear neither in the abstract nor in the results. So I think that the author should make an effort to explain shortly and clearly the steps of the method, and clarify whether the results are divided into "simulations" and "observations" or if there is some combination: e.g., around lines 124-132 simulations are only used but it seems that after line 133 observations are used

I appreciate your attention to detail and for highlighting the need for clarity here. I acknowledge that the organization of the method and results sections could be improved for better understanding in order to be more accessible to readers. I will then take your suggestions into consideration and make the necessary revisions.

In this paper, I present two key developments. Firstly, I propose a representation for the wind-wave coherence spectrum crucial for creating turbulence data to be used in structural load analysis models and also creating synthetic wind-wave interaction data during swell wave conditions. Secondly, I introduce a method to separate wind and wave components using only sonic anemometer high frequency observational data. This method is implemented for both the synthetic model data and observational data. In the light of these two developments and along with your comment, following changes are made:

- I will revise the abstract and introduction to provide a more concise and clear overview of the method's steps, ensuring that the distinction between 'simulations' and 'observations' is explicitly stated.
- In the results section, I will modify further to provide a step-by-step explanation of the method, clarifying how simulations and observations

are utilized, and where any combinations occur (I will add two extra figures to address this concern).

- Specifically, I will revise lines 124-132 (and this section in general) to provide a clear explanation of the use of simulations and ensure that the transition to the use of observations after line 133 is seamless and well-explained.

To bridge between the theoretical model and observation, I will add one figure by including two sonics data. This helps further clarification of method.

3. The decomposition you are presenting in Eqn. (1) is generally known as "triple decomposition" (see e.g., Buckley and Veron, 2017). As most people working in air-sea interaction perform a decomposition like that in the latter study, it would be nice you describe what the differences are between yours and their type of decomposition. Also why not use their type of decomposition?

Various methods exist for decomposing wind-wave interactions, including phase averaging, linear transformation, and orthogonal projection of the wind onto the Hilbert space to estimate the wind-wave coherence signal, etc. Many of these techniques rely on complex cross-spectra between horizontal u and v fluctuating air velocities and vertical w fluctuating air velocities, along with sea surface elevation, to isolate the direct wave influence.

The choice of decomposition method in this manuscript, as outlined in the methodology section, is based on specific considerations related to the research objectives and the nature of the data I am working with (i.e. sonic anemometer data at 15m height above the mean sea level). I plan further to add a figure comparing between the suggested method in this manuscript with one or 2 other decomposition methods. My approach differs in the following ways that I will clarify in the manuscript:

- In summary, the approach solely utilizes sonic wind velocity data, omitting the need for concurrent high-frequency wave measurements in the decomposition process. It neglects velocity fluctuations within the wave band, assuming turbulence field stability during transformation into wavenumber space.

- Additionally, the method stands out as a physics-informed statistical approach that employs a turbulence spectrum model to effectively bridge the gap between high- and low-frequency sections in the observed spectra. This enables us to estimate the variance attributed to turbulent velocity fluctuations within the wave frequency band by learning solely from the energy spectrum of the corresponding wind component.
- Notably, this method uniquely provides wind-corrected and wave time series, a critical data component for structural analysis that is not accessible through other known methods in my knowledge (the link to structural response is an ongoing almost completed work that I am planning to submit).

In short, the decision to employ this decomposition method is rooted in the specific nature of the datasets I am working with, and my extensive experience in motion compensation of moving sensors in both atmospheric and oceanic environments in the presence of a wavy air-sea interface. Through this experience, I have found that filling spectral gaps using a well-established spectrum is an effective approach across a broad spectrum of atmospheric stability and sea state conditions, on both sides of the sea surface.

**To enhance clarity, I will include pseudocode for each method in the manuscript.**

4. In section 2.1 it is not quite clear why you start with a Kaimal wavenumber spectrum and not with a Kaimal frequency-based spectrum, which is much more known and popular.

I acknowledge that the majority of the analysis in this manuscript relies on the frequency spectrum. However, in the "Wind-wave decomposition" section, I incorporate the wavenumber spectrum. Although the Kaimal frequency-based spectrum is widely recognized and used, my choice of the wavenumber spectrum aligns with my approach to solving this specific problem (all my codes on this problem have been developed over time based on wavenumber spectrum). This approach draws inspiration from the one-dimensional wavenumber spectrum of

turbulence as described by Hannoun et al. (1988), Kaimal et al. (1972), and Fung et al. (1992).

5. Also you mention that $\sigma_\beta$ is an adjustable parameter, but is it? Is it not the standard deviation of the variable? If so then it is not adjustable but computable from the time/spatial series. The one that is adjustable as $k_0\beta$ should be A, or am I missing something?

In the manuscript $k_{0\beta}$ represents the spectral roll-off wavenumber for the *\beta* component of velocity. I employed a two-parameter least squares fit of this model spectrum to our observations to allow estimation of $k_{0\beta}$ and $\sigma_{\beta}$, which describe the variance and the spatial scale of the energy-containing eddies. I will try to clarify this further in the manuscript.

6. (7): it is not clear if this is a suggestion made by you based on something or if it is already in the literature. It kind of comes suddenly and you need to provide a background for it.

I appreciate your point about the need for background on the suggestion made for the theoretical coherence function. In general, this study aims for two primary objectives to develop:

- a novel wind-wave coherence spectrum representation, pivotal for generating turbulence data in structural load analysis models and synthesizing wind-wave interactions during swell wave conditions.
- a method for distinguishing wind and wave components solely based on high-frequency sonic anemometer observations or synthetically generated data. This method is successfully applied to both synthetic model data and observational data.

The idea presented in the manuscript is the suggestion made by the author based naturally on prior existing research in the field of wind coherence. Building on these foundations, I proposed this theoretical relationship for the wind and wave

coherence. To enhance the clarity and provide better context, I will include a more explicit reference to the relevant literature and explain a little more the concept.

7. Section 3.1. It is important to state which sonic anemometers are you using, I mean which type of sonics. You mentioned you do not filter for mast shadow (by the way you do not mention which is the direction of orientation of the sonics) but are you using sonic-specific corrections for probe distortions? If so, please tell us which.

As the manuscript is reformulated from letter to regular research, I found this comment indeed important to be addressed appropriately. During the OBLEX-F1 campaign, we deployed two sonic anemometers at the FINO1 offshore meteorological mast. These sonic anemometers were positioned at heights of 15 and 20 meters above mean sea level, with a measurement frequency of 25 Hz. Their orientation was set at 135 degrees, which means that the wind shadow zone extended approximately above 300 degrees. I will include a detailed explanation on this matter in section "3.1 Datasets". I may add a new figure.

**Specific comments**

I have carefully reviewed all of the specific comments raised by the reviewer and they will be precisely incorporated into the revised version of the manuscript. Additionally, I include the reference you suggested.

Buckley M.P. and Veron F. (2017) Airflow measurements at a wavy air-water interface using PIV and LIF. Exp. Fluids. 58:161

**References**

Kaimal, J., J. C. Wyngaard, Y. Izumi, and O. R. Cote, 1972: Spectral characteristics of surface-layer turbulence. Quart. J. Roy. Meteor. Soc., 98, 563–589.

Fung, J. C. H., J. C. R. Hunt, N. A. Malik, and R. J. Perkins, 1992: Kinematic simulation of homogeneous turbulence by unsteady random Fourier modes. J. Fluid Mech., 236, 281–318

Hannoun, I. A., H. J. S. Fernando, and E. J. List, 1988: Turbulence structure near a sharp density interface. J. Fluid Mech., 180, 189–209.

---

## Author Comment (AC2)

Reply to reviewer: for "**Brief communication: Impact of swell waves on atmospheric surface turbulence: A wave-turbulence decomposition method**"

Reviewer comments are presented in black text using the "Calibri" font format with a size of 12.
My responses are displayed in blue text using the "Calibri" font format with a size of 13.

**Reviewer 2**

Wave-Turbulence Decomposition holds significance for both the wave and atmospheric communities. However, it is a considerable challenge for the decompositon. Within this study, the author introduces a method for decomposing wave and turbulence fluctuations. The concept is intriguing and certainly warrants publication. Prior to the manuscript's publication, I have outlined several comments that the author may wish to consider addressing.

Thank for your thoughtful feedback. I'm committed to addressing these invaluable comments to enhance the clarity and quality of the manuscript. In particular, the brief definition of the wave boundary layer is provided to clarify how wave-induced orbital velocities influence velocity fluctuations in the lower marine atmospheric boundary layer, especially in stable conditions. The suggested decomposition method is compared to three wave-turbulence decomposition methods that rely solely on the spectral information of the timeseries. I will elaborate on the observational coherence by available sonic data at two different heights (with separation distance of 5m – I specifically use our sonic data at 20m for this purpose). This study also introduces a theoretical coherence method crucial for generating the cross-spectral density matrix to simulate fluctuating wind velocity and wave height. These contributions emphasize the model's importance and its effectiveness in analyzing aerodynamic and structural loads on offshore wind turbines.

**Specific comments**

1. Several established methods for wave-turbulence decomposition have been utilized in previous studies. To provide context, I recommend that the author furnish an overview of these methods in the introduction. This should encompass the work conducted by Hristov et al. (2003, 2014), the spectral method outlined by Veron et al. (2008) and Grare et al. (2013), as well as the interpolation method described by Rieder and Smith (1998) and Högström et al. (2015).

   I appreciate this feedback. Your suggestion to include references to established wave-turbulence decomposition methods such as those by Hristov et al. (2003, 2014), the spectral method by Veron et al. (2008) and Grare et al. (2013), as well as the interpolation method outlined by Rieder and Smith (1998) and Högström et al. (2015) will enhance the context and comprehensiveness of the manuscript. In response to another reviewer's comment regarding this manuscript's suitability for being considered as a regular manuscript, I have room for further elaborations and will improve the introduction section to provide a more detailed overview and background on the methods (by including references and background). Additionally, I'll include an appendix to explain a few decomposition methods with a figure to compare them with the suggested spectral method of this manuscript, ensuring a more thorough description of the methodology and its effectiveness.

2. In light of these established techniques, it would be beneficial for the author to address whether their proposed method has been compared to these prior approaches. Specifically, have the results obtained using the author's method demonstrated good agreement with those generated by the aforementioned methods?

   In my research, such processing tool, and its application, has evolved over several years, involving the application of various decomposition methods to high-frequency oceanic and atmospheric sensor data, such as shear probes data and ADVs for oceanic studies and sonic anemometer data for atmospheric studies. Notably, these decomposition methods differ, with some relying also on high-frequency surface wave elevation data (beside to wind and current time series) and others solely on high-frequency wind (current) velocity time series. I have conducted thorough comparisons between the suggested spectral method and a

couple of established approaches. These comparisons consistently demonstrated promising agreement, with the spectral method standing out as robust and efficient, acting as a statistical physics-informed gap-filling technique.

It is noted that selecting the appropriate decomposition method can be a complex task, depending on factors such as data characteristics, the nature of wind-wave interaction (like misalignment, stability conditions, etc), advection of wave orbital velocities across a broad frequency range, and more [1]. I've found that the spectral method and its associated technique for deriving wave and corrected turbulence time series from wind speed (ocean current) frequency data are robust and valuable for various atmospheric flux studies over and under the wavy air-sea interface.

I will clearly present and discuss in the manuscript, affirming the effectiveness of the approaches, by comparing the technique with few other ones I made using solely the spectral information, specifically I add an appendix and detail them there.

3. Furthermore, it would be pertinent to explore the strengths and limitations of the author's method in comparison to the existing alternatives. Within the scope of this study, a comparison has been made between the method developed by Hristov and the approach outlined by Veron et al. in 2008. It is noted that the results obtained from these two methods show an acceptable level of agreement, as previously established by Wu et al. in 2008. This comparison adds credibility to the validity of the author's method. Incorporating this comparative analysis would provide a more comprehensive understanding of the novelty and effectiveness of the author's proposed approach in relation to the existing methodologies.

Various methods exist for decomposing wind-wave interactions, including phase averaging, linear transformation, and orthogonal projection of the wind onto the Hilbert space to estimate the wind-wave coherence signal, etc. Many of these techniques rely on complex cross-spectra between horizontal u and v fluctuating air velocities and vertical w fluctuating air velocities, along with sea surface elevation, to isolate the direct wave influence.

The choice of decomposition method in this manuscript, as outlined in the methodology section, is based on specific considerations related to the research

objectives and the nature of the data I am working with (i.e. sonic anemometer data at 15m height above the mean sea level). I plan further to add a figure comparing between the suggested method in this manuscript with one or 2 other decomposition methods. My approach differs in the following ways that I will clarify in the manuscript:

- the approach solely utilizes sonic wind velocity data, omitting the need for concurrent high-frequency wave measurements in the decomposition process. It neglects velocity fluctuations within the wave band, assuming turbulence field stability during transformation into wavenumber space.
- Additionally, the method stands out as a physics-informed statistical approach that employs a turbulence spectrum model to effectively bridge the gap between high- and low-frequency sections in the observed spectra. This enables us to estimate the variance attributed to turbulent velocity fluctuations within the wave frequency band by learning solely from the energy spectrum of the corresponding wind component.
- Notably, this method uniquely provides wind-corrected and wave time series, a critical data component for structural analysis that is not accessible through other known methods in my knowledge.

In summary, the decision to employ this decomposition method is rooted in the specific nature of the datasets I am working with, and my extensive experience in motion compensation of moving sensors in both atmospheric and oceanic environments in the presence of a wavy air-sea interface. Through this experience, I have found that filling spectral gaps using a well-established spectrum is an effective approach across a broad spectrum of atmospheric stability and sea state conditions, on both sides of the sea surface.

To address this comment, I've added an appendix comparing three methods for isolating wave motions from the vertical wind velocity data. These methods, the stopband filter, the intrinsic mode function, and linear interpolation in frequency domain don't rely on wave elevation time series but solely on sonic data. The stopband filter is employed by knowing the wave peak frequency or dominant wave frequency band. I use an estimation for the frequency band as $0.6f_p$ to $f_p+0.1$. The SB filter method significantly reduces energy within the wave-dominant frequencies, resulting in an associated underestimation of turbulent energy; this is approximately the same for the intrinsic mode functions that can be

further improved while the method alone may not completely eliminate the wave velocities. The linear interpolation in the spectral domain may be sensitive to the choice of the wave frequency band.

4. The presence of multiple layers of sonic sensors introduces an intriguing opportunity for validation. It would be highly compelling to ascertain whether the wave coherence contribution as discussed in Section 2.2 aligns with the findings derived from the methods detailed in Section 2.1 across the various sensor layers. This comparative analysis could yield valuable insights into the consistency and reliability of the outcomes.

Thank you for this insightful comment. The idea of a comparative analysis across the various sensor layers is indeed intriguing. In the revised version, I explore the possibility of aligning the wave coherence contribution discussed in Section 2.2 with the findings derived from the methods detailed in Section 2.1 across these multiple sensor layers. This analysis has the potential to provide valuable insights into the consistency and reliability of method outcomes. Furthermore, I will incorporate time series data from another sonic anemometer operating at a 20-meter height. This addition will allow to estimate the observational coherence between the 15-meter and 20-meter sonic anemometers, bridging the gap between the theoretical coherence function proposed and real-world coherence data.

The following figure illustrates coherent structures at two different heights (15m and 20m). In Fig. 1a, a 20-minute sonic data time series at these heights is displayed, while Fig. 1c shows the observed coherence and the theoretical model results. This also sheds light on why the theoretical coherence formula incorporates the wave-induced bump. To address this concept and establish a connection between the observed coherence and my proposed theoretical formula, I will include the fitting of the theoretical coherence to this data. A more detailed explanation will be provided in the methodology and results sections.

[Figure]

Figure 1. The observational and theoretical coherence representations for two sonic anemometers at 15m and 20m heights.

5. It is not easy to follow the connection between sections 2.1, 2.2, and 2.3. Please consider restructuring it.

Thank you for your suggestion regarding the restructuring of methodology section/subsections. I will improve and enhance the clarity and comprehensiveness of both the introduction and methodology sections of the paper.

In response to restructuring, I will undertake the following steps to address your comments:

**Introduction section:** I will restructure the introduction to provide a more comprehensive overview of the established methods for wave-turbulence

decomposition. This will include references to the work by Hristov et al. (2003, 2014), the spectral method by Veron et al. (2008) and Grare et al. (2013), as well as the interpolation method outlined by Rieder and Smith (1998) and Högström et al. (2015). By enhancing this section, I will offer readers a stronger foundation for understanding the context of this research.

**Methodology section:** I will revise the methodology section to explicitly address the comparison between the proposed methods and the established approaches (I use specifically three wave-turbulence decomposition methods as explained in reply to comment 3). This comparison will be presented in a more structured and detailed manner, highlighting the consistency and agreement observed in the evaluations, by emphasizing on the robustness and efficiency of the suggested spectral method as a statistical physics-informed gap-filling technique.

**Result section:** I will correspondingly revise the result section by adding two new figures (Figure 1 is a sample plot).

I believe that these changes will significantly enhance the quality and clarity of the work.

6. For section 2.3: Eq. 11 is only valid for the surface. Thus, it should not have the dependent on z which is confused the readers.

I agree and the text is enhanced by defining the wave boundary layer as the region where the non-static pressure distribution on the surface layer becomes apparent, with a height of impact corresponding to several significant wave heights (Hs). For medium waves, the typical WBL height is a few meters, while for larger waves, it can extend up to say 20 meters. The WBL interacts with the wave surface below and merges with the Monin-Obukhov stratified boundary layer above. Within the WBL, surface wave movements influence the structure, which is shaped by the specific characteristics of the wave field.

7. In the manuscript, you use many "we". Since there is only one author, it should be "I" instead.

Thank you for your feedback. The use of "we" in my manuscript is my impression that this is a common convention in academic writing, even when there is a single author. It can make the writing more formal and objective. However, I can certainly make the change to use "I" instead if it is the preferred style.

**References:**

[1] Bakhoday Paskyabi, Mostafa. A wavelet-entropy based segmentation of turbulence measurements from a moored shear probe near the wavy sea surface. *Springer Nature Applied Sciences* 2019.

---

## Author Response (AR2)

Reply to reviewer's comments for "**Impact of swell waves on atmospheric surface turbulence: Wave-turbulence decomposition methods**"

Reviewer comments are presented in black text using the "Calibri" font format with a size of 12. My responses are displayed in blue text using the "Calibri" font format with a size of 13. The text in the paper are shown as orange text in the "Calibri" font format with a size of 13.

**Comments**

its actual state reads quite unorganized and it needs some organization to make the readers follow precisely the main innovations, goals and the validation/evaluation of what you propose. Therefore, I think the manuscript still needs several major revisions. My comments are based on the annotated/trach changes version of your manuscript.

1. As commented before, the paper lacks some clear organization and is difficult to follow. The results section, as an example, appears as an intermittent provider of results and evaluations.

I appreciate the reviewer for dedicating time to provide insightful comments, and I largely see them helping better clarity of the work. I have tried to address each comment.

In the results section, I added a brief introductory phrase and have included headings to categorize the results based on different methods, as given, to some extent, in the methods section.

**4  Results**

In this section, I utilize three-dimensional wind speed for calculating measured turbulence and wave-induced stresses. For clarity and brevity, the vertical wind component is used to study the performance of the wave-turbulence decomposition techniques.

205  This further provides insights into vertical motion relevant for studying vertical momentum transport, vertical coherence, and turbulence variation with height. I initiate this section with idealized examples, based on the proposed coherence model, to establish a foundational understanding of the developed techniques under controlled parametric conditions.

The results section has been divided into 4 subsection to assure clarity and clear structure: 4.1 Theoretical coherence; 4.2 Measured wind-wave spectra; 4.3 Spectral and coherence analyses; and 4.4 Momentum flux estimation

1.1 Even at the abstract level it is difficult to see the organization of the work: "The primary goal of this study…" sentence, as an example, appears kind of suddenly. Maybe this should be further up in the abstract?

This part has been fully revisited to avoid any potential confusion as follows:

"To study turbulence properties, specifically vertical momentum fluxes during swell wave conditions, we investigate the impact of waves on the power spectrum and spectral coherence of turbulent wind across various spatial and temporal scales. Using a theoretical model derived from sonic anemometer measurements at heights of $15$ m and $20$ m above the mean sea level, …"

1.2 Also in line 11 you stated "two days of sonic anemometers" which gives the impression you analyzed two full days of high frequency sonic measurements: however, you show in many plots (Figs. 2, 5, 6 and 8) more than two days of measurements, but you also focused on two episodes/events on one single day (as far as I understood), i.e. two ten minutes (are they actually two 10-min events only?) in June 24th. So, is it really just two events that are analyzed, I mean in terms of velocity spectra and coherence? If so this should just be clearer from the start (Abstract and Section 3:Data).

I utilized nearly two days evaluating the overall performance of the methods, more specifically both wave decomposition from wind and the estimation of wave-induced momentum. I have also focused on selected episodes within the study period to highlight the performance and efficiency of the decomposition method alone. The initial section illustrates the method's application over two days, as depicted in Figures 2, 5, 6, and 8. Corresponding changes have been made in the abstract to highlight this.

Using a few days of sonic anemometer wind measurements at $15$ m height from June $20$ to $26$, $2015$, the upward momentum transfer could be observed during high-steady ($\sim 7$ m/s) and decaying wind conditions. During the high and decaying winds, the atmospheric stability changes between unstable and stable conditions, blurring the wave signals due to the thermally/mechanically generated turbulence.

Here, I elaborate on the selected cases, serving as benchmarks, to offer additional insights into the spectral characteristics and the effectiveness of the proposed decomposition technique.

"The vertical wind spectra from selected episodes within the study period, serving as benchmarks, provide insights into the nature of impacts on energy elevation within the wave band during low winds, old sea, and stable boundary layer conditions. These spectra also facilitate an effective performance assessment of the proposed decomposition method. "

1.3 Further, in Fig. 4 you analyze two other events?

This is addressed in my previous response. The selected episodes during the study period are utilized to explore the spectral nature of this interaction in more detail.

2. Line 10 and the implications of this: don't you need information on the three velocity components together with (at least) a coherence model to generate turbulent time series?

It is noted that this method is equally applicable to all three wind velocity components, and we are relying on observed data. Generating 3D synthetic turbulence goes beyond the scope of this study, which may (or may not depending on the selected) require the three velocity components. Here, 1D turbulence time series generator is sufficient to address the main objective as we are not generating a turbulence box for the structural load analysis. A forthcoming manuscript, to be submitted soon, will utilize the theoretical model to generate a time series of turbulence for all three wind components using NREL TurbSIM (constraint turbulence). These time series will then be fed into the openFAST model to examine aerodynamics and structural responses of both bottom-fixed and floating turbines. In the outlined procedure, observational data is utilized, and prior to applying the constrained turbulence tool, we incorporate a decomposition step. It's important to emphasize that the turbulence box, particularly the constrained synthetic turbulence box, is not the focus of this paper.

The sentence reads as you only need a coherence model (which can be independent from velocity spectra models).

The decomposition method employed in this study allows for individual application to each wind speed component—thereby eliminating the necessity to concurrently process all three-dimensional components (because we are going to apply it to our sonic measurements and not generating turbulence box synthetic data for the load problem). This flexibility enables a focused and efficient analysis of turbulence and wave characteristics in specific directions. In section 2.2, I have however applied the decomposition separately to all three components of wind speed.

Note that the theoretical model was fitted to the measured coherence spectrum to distinguish the velocity spectrum into its wave and turbulent components, as depicted in Fig. 6, and Appendix C. It can be independently applied to both horizontal (for u- and v-component) and vertical components of wind.

And the synthetic generation of turbulent fields is an important part of your method and so it should be mentioned in the abstract just after you state that you fit the suggested velocity spectrum to the observed one over the range of wave-affected frequencies: so lines 8—11 should be rephrased to reflect this

The primary objectives, as highlighted in the initial lines of the abstract, center around investigating the influence of waves on the power spectrum and spectral coherence of turbulence.

"….we investigate the impact of waves on the power spectrum and spectral coherence of turbulent wind across various spatial and temporal scales …"

While the generation of 3D turbulence is intricately linked to its applications in load analysis, this section specifically emphasizes the method's representation. It's noteworthy that, for calculating the observed wave-induced momentum flux, the decomposition is applied to all three velocity components. This addition is outlined in line 268.

"... Moreover, we apply the decomposition methods outlined in Section 2.1 to all three components of measured wind velocities to estimate the observed wave-induced $\tilde \tau$. …."

where $E_{ww}^o$ and $E_{ww}^m$ are the sonic based energy spectrum and the model spectrum of the vertical wind speed given by Eq.
280   (3), respectively. Figure 8a shows a measure to assess the strength of wavy structures in the observed velocity spectra. I apply the decomposed turbulence time series when $R > \bar{R}$ (i.e. when there exists a well-pronounced energy elevation around $f_p$). Here, $\bar{R}$ represents the average value of $R$. In Fig. 8b, I compare the total wind stress at the surface obtained from Eq. (13), black curve, with the bulk estimation (red curve) derived from COARE3.6 algorithm (see Eq. 2). The two stress estimates align consistently when $R < \bar{R}$. Additionally, it is noteworthy that the wave-induced form stress at the surface, shown by blue
285   markers, undergoes transitions from positive to negative for swells moving opposite to the wind direction (i.e., when $R > \bar{R}$). Figure 8c shows that the estimated form stress at $z = 15$ m according to Eq. (14) is approximately in acceptable agreement with the measured $|\tilde{\tau}|$ from the sonic data, using eddy covariance technique according to Eq. (2). In estimated form stress at 15 m height, the dimensionless function for the vertical decay, i.e. Eq. (14), plays a significant role in vertical distribution of the wave-induced momentum flux. Moreover, the decomposition methods outlined in Section 2.1 have been applied to all
290   three components of measured wind velocities to estimate the observed wave-induced stress $|\tilde{\tau}|$ represented in Eq. (2). Figure 8d illustrates that the ratio of wave-induced (turbulence) intensity (the standard deviation of $\tilde{w}$ over the mean wind speed) to corrected turbulence intensity (the standard deviation of the corrected $w$ over the mean wind speed) is most pronounced when wave elevations are clearly visible around the peak frequency $f_p$ (in agreement with Fig. 8a).

I modified very briefly the methodology section as follows to highlight further where I use three dimensional wind components.

at the flow speed from those traveling at the wave speed (Ayet and Chapron, 2022). This multiscale wind-wave coupling,

65  mediated by wave-coherent motions, is a responsible mechanism for variations of turbulent characteristics over the swell waves. For instance using Eq. (1), the total wind stress vector over the wavy surface is given as follows

$$\boldsymbol{\tau} = \boldsymbol{\tau}' + \tilde{\boldsymbol{\tau}}, \tag{2}$$

where $\boldsymbol{\tau}' = -\rho_a(\overline{u'w'}, \overline{v'w'})$ is the turbulent stress and $\tilde{\boldsymbol{\tau}} = -\rho_a(\overline{\tilde{u}\tilde{w}}, \overline{\tilde{v}\tilde{w}})$ denotes the wave-induced stress. Here, $\rho_a$ indicates the air density. The total wind stress in Eq. (2) can be determined either through high-frequency measurements using the

70  eddy covariance technique, to calculate the observed $\boldsymbol{\tau}'$ and $\tilde{\boldsymbol{\tau}}$, or by employing a bulk formula such as the one provided by COARE3.6 (Edson et al., 2013).

**2.1 Wind-wave decomposition**

3. Following on point 1, I wonder whether section 2.2 (which tells the reader the procedure to separate wind spectrum from wave spectrum) should be before section 2.1 (which introduces a coherence function and the synthetic fields). Then, you could somehow include the time series generation you describe in lines 132-140 in the section with the synthetic generation.

Reviewer's comment has been addressed.

4. Line 123: do you actually need this? I mean can you not just use the relation f S(f) = k F(k)? is the dispersion relation (which you do not describe) really need it?

Yes. I added a new appendix to clarify this

**Appendix A:  Calculation of $df/dk$ based on linear dispersion relation**

The wavenumber and frequency spectra are interrelated through the dispersion relation ( I assume here the linear dispersion relation):

325    $\omega^2 = gk\tanh(kd),$

where $\omega$ is the angular frequency, and $d$ denotes the water depth. The spectral variance, whether expressed in frequency or wavenumber spectra, can be determined accordingly:

$$\sigma^2 = \int F(k)dk = \int E(f)df,$$

where $E(f)$ and $F(k)$ are the frequency and wavenumber spectra respectively. Assuming linear dispersion, we can estimate

330    $dk/df$, essential for the transformation between these two spectra as

$$\frac{df}{dk} = \frac{g}{4\pi\omega}\left[\tanh(kd) + \mathrm{sech}^2(kd)kd\right].$$

5. Following on point 1, I wonder if it helps to have a first subsection in Section 4 where you show the results for the "ideal setup", i.e., Fig. 3a (and somehow it is weird that in the same figure you also have what appears as results from the two episodes you analyzed).

Reviewer's comment has been addressed.

**Minor comments**

1. The use of "we" should be replaced by "I" since I think you did the study alone (at least you wrote "my knowledge" under conclusions)

I tried to address this for the entire paper!

2. Line 24: delete the parenthesis and add "with" before "low"

Reviewer's comment has been addressed.

….Within the WBL, particularly under the influence of swell waves with low to moderate wind speeds, MOST or the logarithmic law …

3. Line 32: you already introduced "MOST" so use instead of "Monin…"

In line 32, I am using "Monin-Obukhov scaling" that cannot be replaced by MOST

4. Line 61 and maybe others: you already introduced WBL so use the acronym
Reviewer's comment has been addressed.

… in this section the WBL through  …

5. Lines 67-68: something is grammatically wrong with this sentence… kind of reads as incomplete

The old version is as follows:

As discussed, the undulating ocean surface generates wave-coherent perturbations in the velocity (and pressure) fields, potentially exerting a dominant influence on turbulent properties within the WBL…

I could not identify any apparent grammatical issues. However, for the sake of ensuring clarity for the readers and appreciate reviewer's comment, I have made enhancements to the sentence as follows:

"The undulating surface of the ocean, as previously discussed, produces wave-coherent perturbations in the velocity (and pressure) fields. This has the potential to exert a dominant influence on turbulent properties within the WBL."

6. Line 124: last sentence of that page was already mentioned

Thanks anyway for this reviewer's comment. I prefer to retain this sentence as it emphasizes that curve fitting is conducted both below and above the wave-affected band, enhancing overall clarity.

7. Line 140 "Eqs. 11 and 11"?

I could not identify any issue here, as I mean exactly Eq. (1).

8. Lines 159 and 160 what medium and large waves mean? What are the sizes?

I agree the terms "medium waves" and "large waves" are relative descriptions, and their specific sizes can vary based on the classification system used. Here large waves generally refer to waves that exceed 4m in height and medium waves may fall within a range of approximately between 2-4m in height. I found these representation in line 41 and clarified as follows:

"For medium waves (approximately $2$ to $4$ m in height), the typical WBL height is a few meters, while for larger waves (more than $4$ m in height), it can extend up to say $20$ m."

9. Line 161 you are sure you meant Eq. (1)?

Yes, if you mean line 141.

10. Line 167 do you really mean the wave variance spectrum or just the wave spectrum?

I think of the wave variance spectrum as a specific type of wave spectrum focusing on the variance (or energy content) across frequencies. But the term "wave spectrum" is used in a more general form containing various types of spectral representations used to analyze wave characteristics in different domains (frequency, wavenumber, etc.).

11. Line 174 sentences on form drag should be after Eq. (15) is introduced

I could not identify any issue and think the order of equations sounds ok to me after checking.

12. Line 210 L has units of m I guess

Yes. I modified as follows
"denotes the Obukhov length scale (in meter) …"

13. Eq. (16) should be moved where appropriate under methods

Addressed! It has moved to the end of Section 2.2.

The observed coherence of vertical velocities is determined using the following relationship:

$$\gamma(z_1, z_2, f) = \frac{|Co_{z_1 z_2}(f)|}{\sqrt{E^{z_1}_{w'w'}(f) E^{z_2}_{w'w'}(f)}},$$
(12)

where $E^{z_1}_{w'w'}(f)$ and $E^{z_2}_{w'w'}(f)$ are the the power spectral density at heights $z_1$ and $z_2$, respectively. $Co_{z_1 z_2}(f)$ denotes the two-point cross-power spectral density at heights $z_1$ and $z_2$.

145  **2.3  Air-sea momentum flux**

14. The words "physics-informed" and "learning solely" are very much machine learning jargon. You do not really use machine learning so I would rephrased these sentences to reflect that

I just simply removed this.

15. Line 319 "discussed in a separate independent study" if you cannot reference this then rephrase otherwise why mentioning this at all.

I prefer to keep it as it is and.

16. Fig.1b it looks like both $E_{ww}/E_{uu}$ and $E_{vv}/E_{uu}$ approach the black line of 3/4 but in line 329 3/4 should be the inverse ratio

I think this line is correct and 4/3 is just for better representation.

"It's evident that the non-corrected ratios in Fig. B1b approach a value of 3/4 for frequencies larger that …"

---

## Author Response (AR3)

Reply to reviewer: for "**Impact of swell waves on atmospheric surface turbulence: Wave-turbulence decomposition methods**"

Reviewer comments are presented in black text using the "Calibri" font format with a size of 12.
My responses are displayed in blue text using the "Calibri" font format with a size of 13.

**Reviewer 1**

1. If sigma_beta in Eq. 3 is the standard deviation, there is no reason this is an adjustable parameter. It is a parameter that one could compute. In principle A and k_0beta should be the only adjustable parameters.

This is the variance estimate through fitting by discarding the waveband contribution. The method is based on a two-parameter least-squares fit between the model spectrum and observations to estimate σ and ko, which describe **the variance** and spatial scale of the energy-containing eddies, respectively. The fit has been performed in log-log space to give all parts of the model fit equal weight. This method has also been tested for measuring the dissipation rate of TKE, which is beyond the scope of this paper. The variable is velocity variance, which, in Gerbi et al. (2009), Eqs 16-17, has been used and compared with the estimate of the same quantity suggested by D'Asaro (2001). I then refer to Appendix 3 of

*Bakhoday Paskyabi, Mostafa. A wavelet-entropy based segmentation of turbulence measurements from a moored shear probe near the wavy sea surface. Springer Nature Applied Sciences 2019.*

And also Gerbi et al 2009. I have also added above reference in the description:

where $k$ denotes wavenumber, $\beta = u, v, w$, $A = 5\sin(3\pi/5)/(6\pi)$ is a constant, and $k_{0\beta}$ and $\sigma_\beta$ are two adjustable parameters describing the roll-off wavenumber (the length scales of turbulent eddies in the energy-containing subrange) and the standard deviation of $\beta$, respectively (Gerbi et al., 2009; Bakhoday-Paskyabi, 2019). Here, I perform a two-parameter least squares fit

2. You have some repeated sentences, e.g., those in lines 85-86

Few changes applied to this paragraph. As follows:
- Line 77: are two parameters
- Line 86: I added " to ensure equal weight is given to all parts of the model fit" for the sake of further clarity

In the first part, from line 79-80, the aim is to introduce the fitting parameters. Towards line 86, additional details about the type of fitting and more insights are provided. I have checked and applied the changes according to the reviewer's comment, ensuring that the explanations are consistent and necessary for clarity in describing the curve fitting procedure.

3. Line 88: I think it is nice you are honest about saying that the interval is determined with trial and error, but then what happens for other datasets? Could you say they also need to do this? Or can they use the same interval?

This is a reliable interval for the vast majority of the OBLEX-F1 campaign dataset: I modified accordingly as follows:

"This interval is determined through a trial-and-error process using all sonic datasets employed in this study, providing a reliable estimate of the frequency band for the entire campaign dataset too."

4. Line 90: this sounds as you do two fits. One below waveband and the other above it. Then this means that you might have 4 different parameters (2 per fit) and then what do you do?

In summary, this approach neglects velocity fluctuations only within the observed wave band and applies the fitting based on frequencies below and above the wave frequency band. Therefore, I use only two parameters for curve fitting. I cited another reference here to facilitate the understanding of method (appendix 3).

*Bakhoday Paskyabi, Mostafa.* *A wavelet-entropy based segmentation of turbulence measurements from a moored shear probe near the wavy sea surface. Springer Nature Applied Sciences 2019.*

using all datasets employed in this study, providing a reliable estimate of the frequency band for the entire campaign dataset too. The energy spectrum is then divided into three bands: below-wave-band ($k < 0.6k_p$), wave-band, and above-wave-band ($k > k_p + 0.1$) parts, see Fig. 1 and refer to Appendix 3 of Bakhoday-Paskyabi (2019). After discarding the wave-band, the Kaimal spectrum Eq. (3) is fitted over below and above wave-band wavenumbers and replace the wave-induced bump by the

5. In many instances you have word "cuttings" such as "doesn't" or "It's". This is not scientific language.

Thanks for the comment. It has been applied for the entire paper by changing from "doesn't" to "does not" and "it's" to "it is".

6. I still do not know which type of sonic anemometers you use and if there is any correction to the probe distortion.

Both 15m and 12m, we used a Gill R3-100. In this analysis no correction applied to the probe distortion.

"During the OBLEX-F1 campaign between $2015$ and $2016$ two additional **Gill R3-100** sonic anemometers were installed at 15 and 20 m above"

7. Line 200: L has units

Yes. The L unit is meter and it has been introduced in line 198 (this is why I just used L<0 and L>0 in this line).

8. Line 203: there is no such a thing as a three-dimensional wind speed

I changed to "three-dimensional wind measurement"

9. Fig 2 does not portrait the Obukhov length as the caption says but z/L

Caption of Fig. 2 has changed for "(c) the values of the stability parameter from which the Obukhov length, $L$, was calculated from sonic measurement at height of $15$ m above …."

10. Sentences around line 220 does not read very scientific

I tried to slightly modify this but the essence remains the same to reflect the purpose of plot:

"The parameters depicted in this figure are idealized to enhance conceptual clarity and technical demonstration, rather than reflecting real-world values. They serve as a theoretical framework to facilitate detailed illustrations of the underlying concept and are not representative of practical measurements or actual conditions."

11. Caption fig 3: 0.1 at the end has units

I modified the fig. Caption:

"... Furthermore, the green-coloured areas in these figures represent the wave-affected frequency band with lower and upper frequencies of $f\_l=0.6f\_p$ in Hz and $f\_u=f\_p+0.1$ in Hz, respectively."

12. Fig. 4 for the c frame would not be better to see a frequency pre-multipled spectra that is scaled with wind speed squared for example? Right now spectra beyond 26-05 disappears because wind speeds are too low and the spectra does not have the energy of the period before

Thank you for your comment. In order to maintain consistency in the plot, I attempted various methods to visualize both Fig 4.b and c, but ultimately decided not to make changes for the sake of representation.

13. The L values above Fig. 6 have units?
The figure has been replotted.

---

## Author Response (AR4)

Dear Andrea Hahmann,

Thanks for the comment. I have re-run the latex code and uploaded the correct tracked changes, along with resubmitting the last version (which was correct).